# Neuromorphic overparameterisation and few-shot learning in multilayer physical neural networks

Kilian D. Stenning [1,2] ✉, Jack C. Gartside [1,2,9], Luca Manneschi[3,9], Christopher T. S. Cheung[1], Tony Chen[1], Alex Vanstone [1], Jake Love[4], Holly Holder [1], Francesco Caravelli [5], Hidekazu Kurebayashi [6,7,8], Karin Everschor-Sitte [4], Eleni Vasilaki [3] & Will R. Branford [1,2]

Physical neuromorphic computing, exploiting the complex dynamics of physical systems, has seen rapid advancements in sophistication and performance. Physical reservoir computing, a subset of neuromorphic computing, faces limitations due to its reliance on single systems. This constrains output dimensionality and dynamic range, limiting performance to a narrow range of tasks. Here, we engineer a suite of nanomagnetic array physical reservoirs and interconnect them in parallel and series to create a multilayer neural network architecture. The output of one reservoir is recorded, scaled and virtually fed as input to the next reservoir. This networked approach increases output dimensionality, internal dynamics and computational performance. We demonstrate that a physical neuromorphic system can achieve an overparameterised state, facilitating meta-learning on small training sets and yielding strong performance across a wide range of tasks. Our approach's efficacy is further demonstrated through few-shot learning, where the system rapidly adapts to new tasks.

In artificial intelligence (AI) and machine learning, the performance of models often scales with their size and the number of trainable parameters. Empirical evidence supports the advantages of overparameterised regimes[1,2], where models, despite having a large number of parameters, avoid overfitting, generalise effectively and learn efficiently from a limited number of training examples.

Physical neuromorphic computing aims to offload processing for machine learning problems to the complex dynamics of physical systems[3–15]. Physical computing architectures range from implementations of feed-forward neural networks[15,16] with tuneable internal weights, to reservoir computing, where complex internal system dynamics are leveraged for computation[8]. Neuromorphic schemes stand to benefit from the advantages of operating in an overparameterised regime. A primary use-case of neuromorphic systems is in edge-computing[17] where a remotely situated device locally performs AI-like tasks. For example, an exoplanet rover vehicle performing object classification can operate more efficiently by processing data on-site rather than transmitting large datasets to cloud-servers[7]. Here, acquiring large datasets and transmitting them to cloud servers for processing is often inefficient, making the ability to compute well and adapt to new tasks with small training sets a desirable function.

[1]Blackett Laboratory, Imperial College London, London SW7 2AZ, United Kingdom. [2]London Centre for Nanotechnology, Imperial College London, London SW7 2AZ, United Kingdom. [3]University of Sheffield, Sheffield S10 2TN, United Kingdom. [4]Faculty of Physics and Center for Nanointegration Duisburg-Essen (CENIDE), University of Duisburg-Essen, 47057 Duisburg, Germany. [5]Theoretical Division (T4), Los Alamos National Laboratory, Los Alamos, NM 87545, USA. [6]London Centre for Nanotechnology, University College London, London WC1H 0AH, United Kingdom. [7]Department of Electronic and Electrical Engineering, University College London, London WC1H 0AH, United Kingdom. [8]WPI Advanced Institute for Materials Research, Tohoku University, Sendai, Japan. [9]These authors contributed equally: Jack C. Gartside, Luca Manneschi. ✉e-mail: k.stenning18@imperial.ac.uk

In physical reservoir computing, the internal dynamics of a system are not trained. Instead, only a set of weights to be applied to the readout layer are trained, reducing training costs when compared to neural network architectures. Typically, only a single physical system is employed with a fixed set of internal dynamics, resulting in a lack of versatility and overly specialised computation, which is fixed at the fabrication stage. In contrast, the brain possesses a rich set of internal dynamics, incorporating multiple memory timescales to efficiently process temporal data[18]. To mimic this, research on software-based reservoirs has shown that combining multiple reservoirs with differing internal dynamics in parallel and series network architectures significantly improves performance[19–26]. Parallel networks have been physically implemented[8,27,28], however they lack inter-node connectivity for transferring information between physical systems—limiting performance. Translating series-connected networks (often termed hierarchical or deep) to physical systems is nontrivial and so-far unrealised due to the large number of possible inter-layer configurations and interconnect complexity.

Here, we present solutions to key problems in the physical reservoir computing field: we fabricate three physical nanomagnetic reservoirs with high output dimensionality and show how increasing the complexity of system dynamics can improve computational properties (Nanomagnetic reservoirs, Fig. 1). We then develop and demonstrate a methodology to interconnect arbitrary reservoirs into networks. We demonstrate the computational benefits of the networked architecture and compare performance to software. Reservoir outputs are experimentally measured with network interconnections made virtually and outputs combined offline during training (Multilayer physical neural network, Fig. 2). We explore the overparameterised regime, made possible through the networking approach, where the number of network output channels far exceeds the size of the training set. The physical networks architectures do not overfit, show enhanced computational performance and are capable of fast learning with limited data. This approach is applicable to all physical systems and methods to achieve overparameterisation in an arbitrary system are discussed (Overparameterisation, Fig. 3). We demonstrate the power of operating in an overparameterised regime by implementing a few-shot learning task using model-agnostic meta-learning[29–31]. The physical network is able to rapidly adapt to new tasks with a small number of training data points (Learning in the overparameterised regime, Fig. 4).

Whilst we use nanomagnetic reservoirs to demonstrate the benefits of networking and overparameterisation, the methodology can be applied to any physical system. Additionally, we discuss the scalability of our nanomagnetic computing scheme and calculate the theoretical power consumption of a device. The scheme described here lifts the limitations of low-dimensionality and single physical systems from neuromorphic computing, moving towards a next-generation of versatile computational networks that harness the synergistic strengths of multiple physical systems. All data and code are publicly available[32].

## Results

### Nanomagnetic reservoirs

The physical reservoirs used here are nanomagnetic arrays that have both nonlinearity (NL) and fading memory (i.e., a temporal response that depends on both current and previous inputs)[5,33]. Each array comprises many individual nanomagnetic elements, each with its own magnetisation 'state'. When applying an external input, the elements in an array may switch magnetisation state. The switching process depends on the external input, the current state of an element, and the states of neighbouring elements through dipolar coupling, which gives rise to collective switching and high-frequency dynamics, which we utilise for computation. Here, we drive switching with external magnetic fields. The array state is read by measuring the absorption of injected microwaves (i.e., measuring resonance) using ferromagnetic

resonance spectroscopy (FMR), producing a spectra that is highly correlated with the collective state of the array[5,34–36].

Figure 1 shows scanning electron micrographs (Fig. 1a, c, e) and FMR spectral evolution of three nanomagnetic reservoirs (labelled macrospin (MS), width-modifed (WM)[5] and pinwheel (PW), see Supplementary Note 1 for details) when subject to a sinusoidal field-input (Fig. 1b, d, f). A schematic of the computing scheme is shown in Fig. 1g). The three arrays are designed to produce different and history-dependent responses to explore the effects of networking system with distinct dynamics. MS is a square lattice with bars only supporting macrospin magnetisation states. WM is a width-modified square lattice capable of hosting both macrospin and vortex spin textures. PW is a disordered pinwheel lattice with structural diversity throughout the sample. A detailed discussion of the design is provided in Supplementary Note 1. The high readout dimensionality of FMR is key to achieving both strong performance and overparameterisation. Other techniques for readout exist, such as magnetoresistance[37], albeit at a lower output dimensionality. During training, weights are applied to recorded data offline via ridge-regression, which transforms the FMR spectra to a 1D time series with the aim of closely matching the target waveform (see Methods for further details).

We assess reservoir performance via the mean-squared error (MSE) between the target and the reservoir prediction as well as two metrics: memory-capacity (MC) and NL. MC measures the ability of the current state to recall previous inputs[38], which arises from history-dependent state evolution. NL measures how well past inputs can be linearly mapped to the current state[38], and can arise from a number of physical system dynamics. In this work, these dynamics include resonant frequencies shifts from changing microstates and input field and the shape of FMR peaks. Metric calculations are further described in the Methods section, Task Selection, and Supplementary Note 2. These metrics allow a coarse mapping between physical and computational properties, enabling comparison of different systems. Figure 1h) shows the NL and MC metrics. Our array designs produce a diverse set of metrics, ideal for exploring the computational benefits of networking later in this work.

Throughout this work, we focus on two input time-series: a sine-wave and the chaotic Mackey-Glass time-series[39] (described in the Methods section, Task Selection). These datasets are chosen as they are compatible with 1D global field-input and can be used to devise a number of computational tasks with varying requirements. We employ short training datasets (200 data points) to reflect real-world applications with strict limitations on data collection time and energy.

We wish to evaluate the best possible computation from our readout. To do so, we employ a feature algorithm selecting the best performing combination of readout features (i.e., frequency channels) and discarding noisy or highly correlated features, which do not improve computation (see Methods for details).

Figure 1i shows the MSE of each reservoir when predicting various future values (x-axis) of the Mackey-Glass time-series. These tasks require high memory capacity, as attaining a good prediction requires knowledge of previous inputs, with WM and PW outperforming MS due to their richer internal switching dynamics (Supplementary Note 1). All arrays exhibit performance breakdown at longer future steps, evidenced by the periodic MSE profiles in Fig. 1i). This is clear for the $t$+7 task taken from the high-MSE region (Fig. 1j), where predictions do not resemble the target. At later future steps, prediction quality improves due to the quasi-periodicity of the Mackey-Glass equation ($t$+11 is similar to t).

This breakdown in performance is common in single physical systems and software reservoirs, which often do not possess the range of dynamic timescales (e.g., retaining enough information about previous inputs) to accurately predict future steps and provide a true prediction where performance gradually decreases when predicting further into the future[20,26,40].

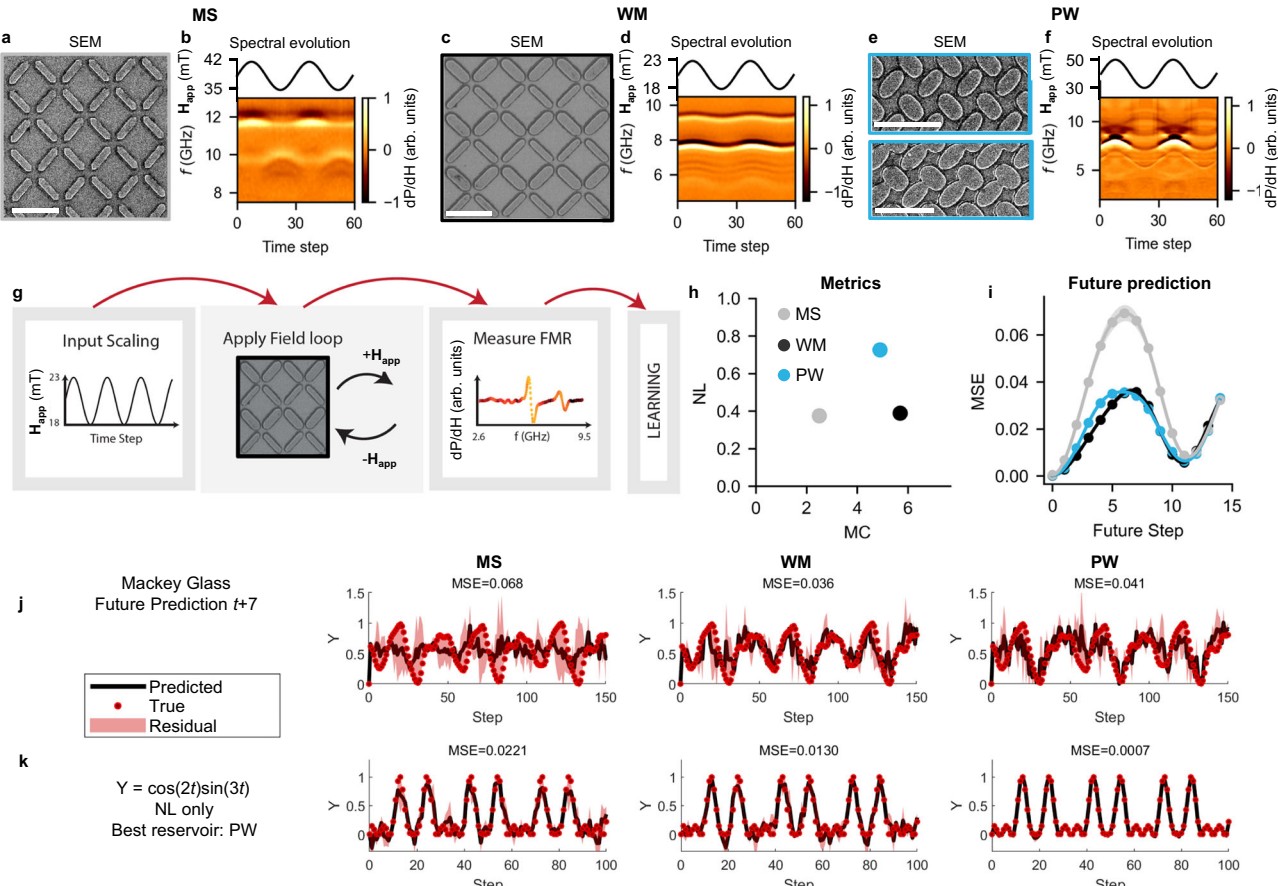

**Fig. 1 | Nanomagnetic reservoirs. a, b** Square macrospin-only artificial spin-ice (MS). **c, d** Width-modified square artificial spin-vortex ice (WM). **e, f** Disorderded pinwheel artificial spin-vortex ice (PW). Bars vary from fully disconnected to partially connected. **a, c, e** Scanning electron micrographs (SEM). All scale bars correspond to 1 μm. **b, d, f** Ferromagnetic resonance spectroscopy (FMR) spectral evolution from a sinusoidal field-series input. Scale bar represents amplitude of FMR signal dP/dH (arb. units). **g** Reservoir computing schematic. Data are applied via magnetic field loops (+$H_{app}$ then −$H_{app}$), which leads to collective switching dynamics in the array. FMR output spectra measured at −$H_{app}$ is used as computational output. **h** Memory-capacity (MC) and nonlinearity (NL) of the reservoirs. Variations in sample design produce a diverse set of metrics. **i** Mean-squared error (MSE) when predicting future values of the Mackey-Glass time-series. High memory-capacity and low nonlinearity (WM) gives best performance. Shading is the standard deviation of the prediction of 10 feature selection trials. **j** Attempted predictions of $t+7$ of the Mackey-Glass equation. No single reservoir performs well. **k** Transforming a sine-wave to $\cos(2t)\sin(3t)$. PW has 31.6 × lower MSE than MS. Shading represents the residual of the prediction.

Figure 1k shows the performance when transforming a sine-wave to $\cos(2t)\sin(3t)$. The disordered PW array achieves up to 31.6× improvement for transformation tasks with no additional energy cost, highlighting the significance of system geometry and design. Performance for a history-dependent non-linear transform task (non-linear autoregressive moving average, NARMA, transform[41]) and further sine-transformations are shown in Supplementary Note 3. NARMA transforms display similar periodic profiles. For sine-transformations, no one array performs the best at all tasks, further highlighting the lack of versatility of single physical systems.

## Multilayer physical neural network

To overcome the limitations of single physical systems, we now interconnect individual reservoirs to form networks, where each reservoir acts as a complex node with memory, NL, and high output dimensionality (as opposed to traditional neural network nodes, which have no memory and only 1 output dimension).

We begin by making parallel and 1D series networks. In parallel networks, inputs and readout for each array are performed independently and combined offline (Fig. 2a). In series networks, the output from one node is used as input to the next (Fig. 2b). In this work, interconnections are made virtually, as opposed to real-time interconnection. We experimentally record all data from the first node and

select one output channel to be used as input to the next node. After scaling to an appropriate field range, this time-series is then passed to the next node and its response is recorded. The full readout from both layers is then combined offline during training and prediction. We trial 49 total architectures (4 parallel and 36 2-series and 9 3-series networks with different configurations of arrays interconnections). Full details of each architecture can be found in the supplementary information. We combine the outputs of every series/parallel architecture offline to create a network that is analogous to a physical neural network (PNN) (Fig. 2c).

As each node has ~200 readout channels, the optimisation time required to evaluate all interconnections is not practically feasible. Seeking a more efficient solution, we explore how the output-channel characteristics from one node affect the MC and NL of the next node in a series network. Figure 2d, e show the per-channel NL and correlation (Corr) to previous inputs values for WM (see Methods for calculation details). Certain frequency channels are highly correlated with previous inputs (e.g., 7.9−8.1 GHz), whereas others are highly non-linear (e.g., 7 GHz). By feeding the output of one channel as input to the next node we can evaluate how output characteristics affect the next node's MC and NL. Figure 2f shows this relationship when the second node is WM (other architectures shown in Supplementary Note 4). $Corr_{in}$ and $NL_{in}$ refer to the correlation and NL of the first node output channel.

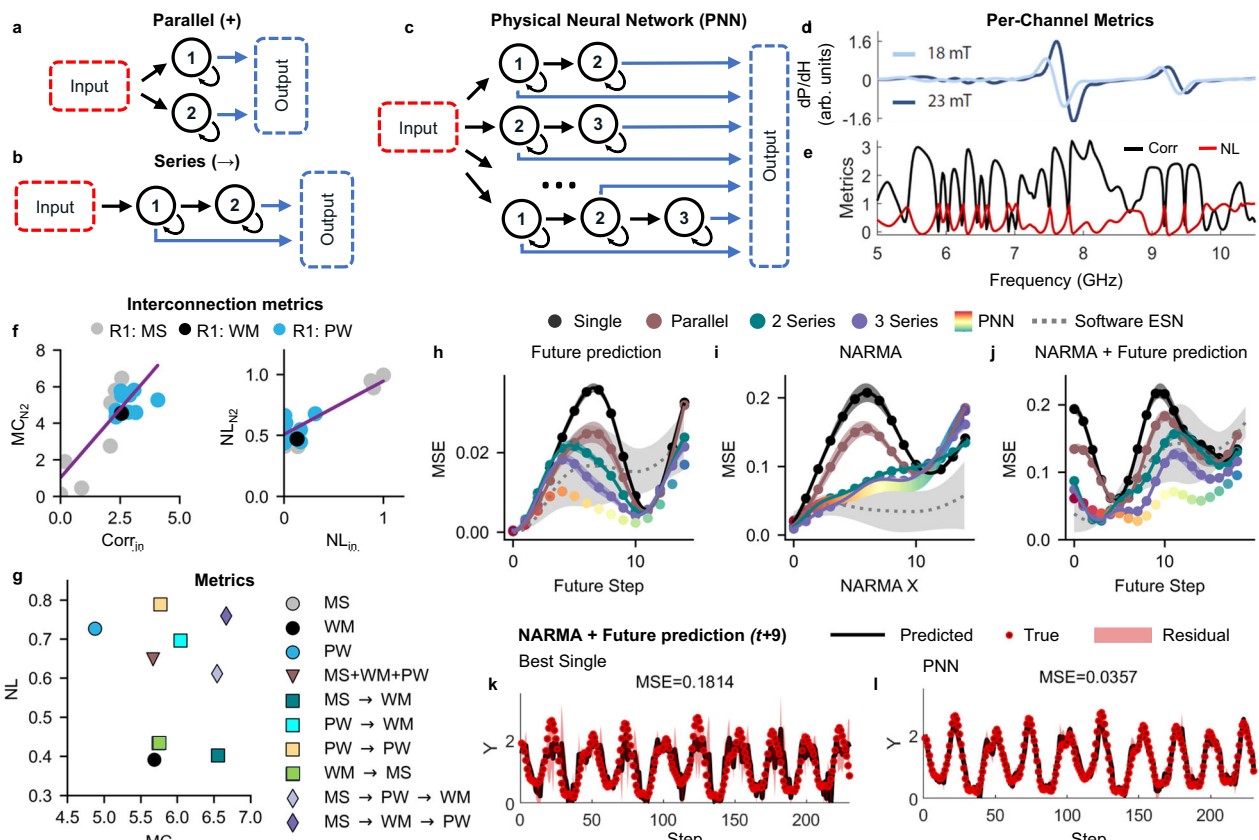

**Fig. 2 | Multilayer physical neural networks with complex nodes.** Schematics of **a** parallel (+) networks, **b** series (→) networks and **c** physical neural networks (PNN). Network nodes are recurrent, non-linear nanomagnetic reservoirs with high output dimensionality. In parallel networks, input is fed to multiple nodes independently. In series, the output of one node is virtually fed as input to the next. PNN networks combine series networks in parallel. All interconnections are made virtually, as opposed to physical interconnection. The response of every node is combined offline to create the output of a network. **d** WM FMR amplitudes at maximum (dark blue) and minimum (light blue) input fields. **e** Frequency-channel signal correlation (Corr) to previous time steps and nonlinearity (NL). These metrics are used to guide which frequency channel is used as input for the next node in a series network. **f** Relationship between first reservoir node output metrics, correlation ($Corr_{in}$) and nonlinearity ($NL_{in}$), used for interconnections and measured second reservoir metrics, memory-capacity ($MC_{R2}$) and nonlinearity ($NL_{R2}$), when the second reservoir (R2) is WM. Lines represent linear fits. Higher metric scores are correlated with higher-scoring reservoir output channels. **g** Memory capacity (MC) and non-linearity (NL) of selected single (circles), parallel (triangles) and series networks (squares for series length 2, diamonds for series length 3). Networks have a broad enhancement of metrics versus single arrays. PNN's can take any metric combination. MSE profiles for **h** Mackey-Glass future prediction, **i** NARMA transformation and **j** future prediction of NARMA-7 processed Mackey-Glass for the best single (WM), parallel (MS+WM+PW), 2 series (MS→WM), 3 series (MS→WM→PW) and PNN. Also shown is the performance of a software echo-state network with 100 nodes. MSE profiles are significantly flattened for the PNN. Shading is the standard deviation of the prediction of 10 feature selection trials or 10 echo-state networks. **k, l** Example predictions for $t$+9 of the NARMA−7 processed Mackey-Glass signal. Shading represents the residual of the prediction.

$MC_{N2}$ and $NL_{N2}$ are the MC and NL of the second node. Points are colour-coded depending on which array acts as the first node (consistent with previous Figures). Both $MC_{N2}$ and $NL_{N2}$ follow an approximately linear relationship. A linear relationship is also observed when comparing correlations to specific previous inputs (Supplementary Note 5). As such, one may tailor the overall network metrics by selecting output channels with certain characteristics. This interconnection control goes beyond conventional reservoir computing where interconnections are made at random[20], allowing controlled design of network properties.

Fig. 2g shows the MC and NL of selected single, parallel, and series sub-networks. Parallel network NL and MC (triangular markers) lie between the NL and MC from the constituent reservoirs. MC does not increase in parallel as no information is directly transferred between network nodes. Series networks (square and diamond symbols for series network depth of 2 and 3, respectively) show memory and NL improvements above any single reservoir. The MS→PW→WM network (purple diamond) has both high memory and NL. PNN's can be devised to possess any metric combination as it comprises all sub-networks. The ordering of arrays in series networks is important, with memory

enhancement only observed when networks are sequenced from low (first) to high (last) memory (e.g., MS→WM has a larger memory capacity than WM→MS), a phenomenon also seen in software reservoirs[20] and human brains[18]. To date, there has been an open question as to whether physical reservoirs are analogous to software echo-state networks. These results suggest that the two are comparable, and that methods used to improve software echo-state networks can be transferred to physical reservoir computing.

We now evaluate the performance of these networks using the feature selection methodology[20,42] described in the Methods section, Learning Algorithms. Figure 2h–j compares the MSE for the best configuration of each network architecture when predicting future values of the Mackey-Glass equation (h), performing a NARMA transform (i) and combining NARMA transform and prediction (j) (additional architectures and tasks are shown in Supplementary Note 6). Parallel arrays (brown lines) do not show significant MSE reductions for these tasks as there is no information transfer and memory increase in this architecture. 2 and 3 series networks show significant decreases in MSE for all tasks, which improve as the series network is extended. The PNN outperforms other architectures across all future time step prediction

tasks (Fig. 2h, j) with significant MSE vs $t$ flattening, demonstrating higher-quality prediction. This is particularly evident when predicting $t$ +9 of the NARMA-transformed Mackey-Glass equation (Fig. 2k, l) and when reconstructing the Mackey-Glass attractor (Supplementary Note 7). For NARMA transform (Fig. 2h), both the PNN and three-series network show a linear MSE vs $t$ profile, indicative of reaching optimum performance. Additionally, the PNN performs well across a host of non-linear signal transformation tasks (Supplementary Note 6), out-performing other network architectures at 9/20 tasks.

A strength of the PNN is the increased readout dimensionality. The results in Fig. 2h–j use 40−100 outputs for the single, parallel, and series networks and ~13,500 outputs for the PNN. This number of outputs is far greater than the size of the training set, placing the PNN in the overparameterised regime (discussed later). In Supplementary Note 8, we constrain the PNN to have a similar number of outputs to the other architectures and find performance is similar to that of a three-series network. As a comparison, the grey dashed line in h–j represents the average performance of an 50 randomly initialised 100-node echo state networks (ESN) (further software comparison provided in Supplementary Note 9). We find that single arrays are well matched to ESNs with ~20 nodes, and PNNs are well matched to ESNs with 500 nodes. Additionally, we initialise three ESNs with similar characteristics to nanomagnetic arrays and interconnect them. Interconnected ESNs, whilst functionally different, also display improvements during

interconnection, highlighting both the general nature of the technique, and strengthening the analogy between physical reservoirs and software echo-state networks.

## Overparameterisation

We now explore the computational benefits of reaching an over-parameterised regime, where the number of network outputs exceeds the size of the training dataset. To reach overparameterisation there are three key parameters: the size of the length of the training dataset, the number of output parameters, which here refers the number of FMR channels, used during the training phase, and the effective dimensionality of the readout. The last point is critical, as the number of parameters can be arbitrarily increased by increasing the measurement resolution or repeating measurements but doing so results in increasingly correlated outputs, which have no impact during training. Here, we show how engineering a rich readout response can be used to reach overparameterisation in physical systems.

Figure 3a, c shows the train and test MSE for the best networks from each architecture (equivalent to Fig. 2) when varying the number of output parameters for training set lengths of a) 50 and c) 100. Figure 3b, d shows the improvement factors ($min(MSE_{UP})/min(MSE_{OP})$) when comparing the MSEs between the underparameterised (UP) and overparameterised (OP) regime. All architectures beyond single systems show improvements when operating in the overparameterised

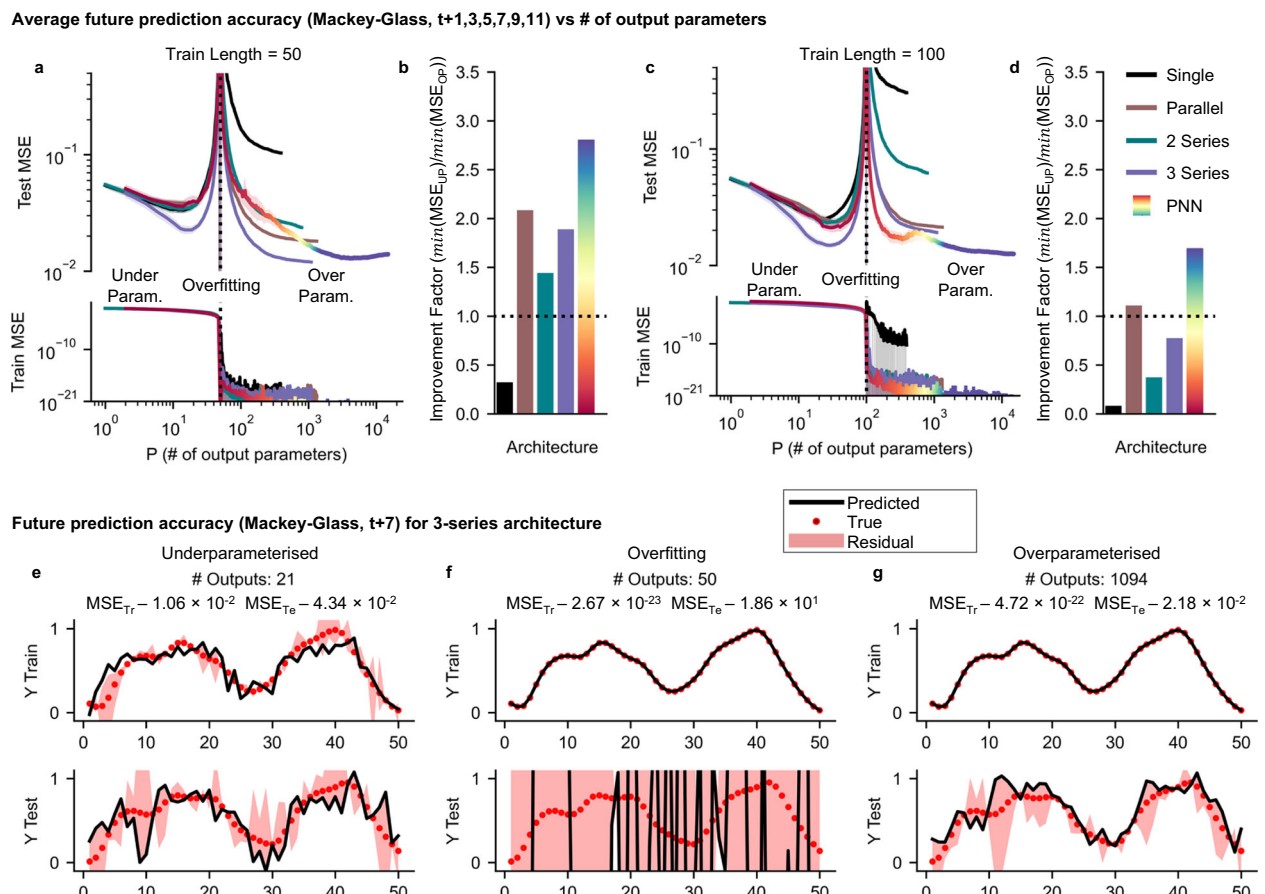

**Fig. 3 | Neuromorphic overparameterisation.** Train and test MSE for selected architectures when predicting future values of the Mackey-Glass equation when varying the number of output parameters $P$ (i.e., FMR output channels) for training set lengths of **a** 50 and **c** 100. MSE is the mean of 50 random combinations of outputs across 6 different prediction tasks (Mackey-Glass, $t$+1, 3, 5, 7, 9, 11). Shading is the standard error of the MSE over 50 trials of randomly selecting features. **b**, **d** Improvement factors ($min(MSE_{UP})$ / $min(MSE_{OP})$) when moving from

underparameterised (UP) regime to overparameterised (OP) regime. Each network shows three regimes: an underparameterised regime when the number of parameters is small, an overfitting regime where the number of parameters is near the size of the training set and an overparameterised regime when the number of parameters exceeds the size of the training set. Example train and test predictions for 3 series architecture in the underparameterised regime (**e**), overfitting regime (**f**) and overparameterised regime (**g**).

regime. Here, MSE is an average of over 6 prediction tasks (Mackey-Glass, $t$+1,3,5,7,9,11) and 50 random trials of adding parameters (see Methods for details).

For each network and training length, we see three regimes with example predictions for each regime shown in Fig. 3e–g) for the 3 series architecture: When the number of output parameters is less than the training length, the system is underparameterised and train and test MSE decrease as the number of parameters increases (Fig. 3e)). This is where the majority of neuromorphic schemes operate. When the number of output parameters is close to the length of the training set, the networks enter an overfitting regime where MSE follows a 'U-trend' and performance deteriorates with increasing number of outputs. Here, there are multiple solutions during training and the network is able to perfectly fit the target, leading to vanishingly small training errors. However, the fitted weights are arbitrary and when presented with unseen data, produce predictions with no resemblance to the target (Fig. 3f). Surprisingly, this trend does not continue as the the number of outputs increases. Instead, the test performance substantially improves. Here, the network enters an overparameterised regime and overcomes overfitting[1,2,43,44]. Instead of memorising the training data, the network can generalise and learn the underlying behaviour of the task, resulting in improved test performance (Fig. 3g). Crucially, for some networks, the overparameterised MSE is lower than the underparameterised MSE. This phenomenon has been observed in software deep learning (sometimes referred to as a double-descent phenomena[43,44]), but not physical computing systems.

The extent to which MSE recovers depends on the effective dimensionality of the readout and train length. For single reservoirs, the limited internal dynamics produce a highly correlated set of outputs. Whilst MSE reduces in the overparameterised regime, the network is unable to fully overcome the effects of overfitting resulting in high overparameterised MSE and improvement factors below 1. The networking approach increases the effective dimensionality of the readout. In parallel networks this is achieved by having distinct nodes with different dynamic responses. In series networks, nodes receive unique inputs that produce different dynamic regimes. For short training lengths (Fig. 3a, b), the enhanced output dimensionality produces a beneficial overparameterised regime with improved MSE. Improvements increase as the network size increases, and even series networks with one node type can achieve a beneficial overparameterised regime (Supplementary Note 10). For longer train lengths (Fig. 3c, d), only the PNN reaches a beneficial overparameterised regime due to the increased effective readout dimensionality required to overcome overfitting. Task also plays a role, with more challenging prediction tasks demanding higher effective readout dimensionality to reach a beneficial overparameterised regime (Supplementary Note 10). Interestingly, the PNN architecture shows signatures of triple descent ($N_{\text{train}} = 50$, $P \sim 150$ and $N_{\text{train}} = 100$, $P \sim 600$) where a peak in MSE is observed in the overparameterised regime, a sign of superabundant overparameterisation[45]. Here we use linear regression to train networks weights. When using gradient-descent, a similar trend is observed showing the results are robust to different training methods (Supplementary Note 11).

These results can be applied to any arbitrary system to improve computation and reach an overparameterised state. To produce a diverse set of outputs, both physical system and readout should be designed in tandem. Readout improvements can be obtained by increasing measurement resolution, fabricating multiple electrodes, or reading at different external biases (e.g., here, we measure FMR at the input field, inducing field-dependent resonant shifts). Moving away from homogeneous systems will produce a greater breadth of internal dynamics, improving performance and enabling a useful overparameterised regime. Examples include introducing structural variation throughout the system (as with the PW array here) or biasing physically separated regions.

The networking approach is an effective method of improving readout dimensionality. Combining distinct systems is beneficial as the breadth of dynamics is likely to be higher, but networking the same physical system can produce overparameterisation. Enriched dynamics and increased effective readout dimensionality can be achieved by operating the physical system in different dynamic regimes, for example, by changing external conditions to access different dynamics[46]. Alternatively, by varying the input sequence, either through a random mask (a common technique known as virtual nodes) or by feeding the output of one node as input to another, a diverse parallel / series network can be created. Even in the limit of a single physical system with one readout per time step, overparameterisation can be achieved through a combination of parallel and series networking, provided that the internal dynamics are rich enough.

### Learning in the overparameterised regime

We now showcase the computational advantages of physical neural networks operating in the overparameterised regime. The high dimensionality and complexity of the network readout permits rapid learning with a limited number of data-points[1,2,31,43,44]. This characteristic is a particularly desirable feature for any neuromorphic computing system as it allows rapid adaptation to changing tasks/environments in remote applications where collecting long training datasets carries a high cost. To demonstrate this, we show a challenging fast few-shot learning adaptation for previously unseen tasks using a model-agnostic meta learning approach[29,31].

Figure 4a shows the system prediction when predicting (left-to-right) the $t + 5$, $t + 14$ and NARMA-processed $t + 14$ Mackey-Glass signals. The PNN is trained on just the first 50 data points of the signal, highlighted by black circles. The PNN is able to learn the underlying system dynamics and provide good predictions demonstrating the power and adaptability of the overparameterised regime. We note that the MSE's achieved are comparable to those in Fig. 2e, g when using feature selection with the PNN. As such, this meta-learning can achieve strong results with a 75% reduction in training set size (50 training data points here vs 200 previously used).

To further showcase the computational capabilities of the PNN, we now demonstrate few-shot learning where the seen training data are sparsely distributed throughout the target dataset, representing, for instance, a very low sampling rate of a physical input sensor in an edge-computing use-case. In Fig. 4b, the system is driven by a sinusoidal input and asked to predict a target of the form $\tilde{y}(t) = \sum_n^{N^\omega} a_n \sin(nt + \theta_n)$ i.e., simultaneously predict amplitude, frequency and phase changes—a task that requires a range of temporal dynamics in the network, often used as a meta-learning benchmark task[31]. The values $a_n$ and $\theta_n$ are sampled randomly at the beginning of each task from continuous uniform distributions (details in the Methods section, Task Selection). The goal is to train the system to generalise to a set of $a_n$ and $\theta_n$ values, and then rapidly adapt to a new task with a limited set of sparse training points.

In all previous tasks, the network is trained to produce a single output. Here, we simultaneously adapt to five distinct functions and sum the predictions to produce the final waveform. This increases the task difficulty as the network must be generalised to all possible amplitude, frequency and phase shifts, and any errors will be amplified in the final output. To achieve this, we use a variation of the MAML meta-learning algorithm[31] applied to the frequency-channel outputs of the network, leaving all history-dependent and non-linear computation to the intrinsic dynamics of the physical network.

Figure 4b, c shows predictions of three example tasks for the overall target and two example sub-components from each task respectively. The network sees just 15 data points (highlighted by grey circles in panel b)) throughout the entire process. In Fig. 4b, the target (dashed black line) and predicted values after updating the generalised

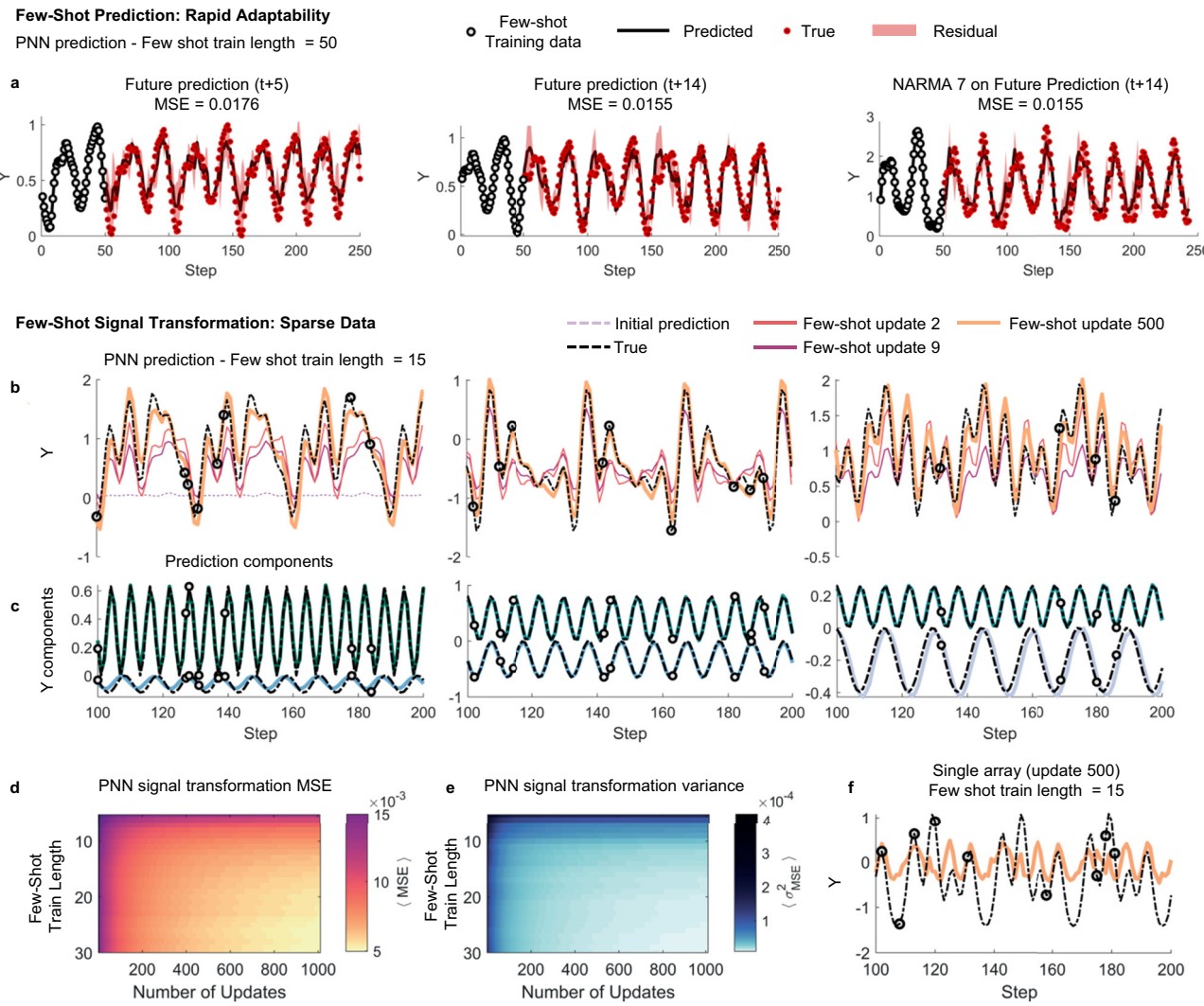

**Fig. 4 | Few-shot learning. a** System predictions (black lines) in the over-parameterised regime when rapidly adapting to 50 training data-points (grey circles) at the beginning of the series. The PNN is able to rapidly learn the task and shows strong performance on the test data set (target points in red). Shading is the residual between true and predicted values. **b** Sine-transformations with sparse training data. The system is asked to learn five different frequency components of a time-varying function (dashed black line). Here, the PNN is shown just 15 sparsely separated training data points (grey circles). As the system is updated (coloured plots), the PNN prediction steadily improves. **c** Two example frequency components of the targets in **b**, which the PNN learns. **d, e** Average error, <MSE>, and variance of the error, $<\sigma^2_{MSE}>$, computed over 500 signal transformation tasks as the train length and updates vary. The system shows good generalisation across all tasks, as shown by the low variance. **f** Few-shot signal transformation using a single array after 500 updates. The single array fails completely.

matrix a number of times are shown. At update 0, the predicted response does not match the target waveform as expected. As the network updates, the error between the prediction and target reduce. Further example tasks are provided in Supplementary Note 12.

Despite the limited information and the high variability of tasks, the network learns the underlying sinusoidal components and can adapt to different targets with the partial information available. To support this claim of generalisation, Fig. 4d, e show the average error <MSE>, and variance of the error, $<\sigma^2_{MSE}>$, respectively, calculated over 500 different tasks for various few-shot train lengths and number of updates. The error decreases as the number of updates and available data points is increased as expected, with strong performance observed for as little as 10 training data points. Crucially, the variance of the error across all tasks is low, demonstrating strong generalisation. Finally, Fig. 4f) reports an example of prediction obtained when meta-learning is applied on a single reservoir, which fails completely at the task.

The meta-learning approach showcases the richness of the high-dimensional output space of the PNN architecture. The network is able to learn the general behaviour and dynamics of the input and a set of tasks enabling rapid adaptability. This removes the requirement for complete retraining, making essential progress toward on-the-fly system reconfiguration.

## Discussion

We have demonstrated how interconnecting physical systems into larger networks can improve computational performance across a broad task set as well as enhance task-agnostic computational metrics. We now discuss the challenges which must be addressed before the PNN architecture can be realised at a future device level, as well as the prospects for scaling nanomagnetic hardware.

Currently, the PNN is created offline, with the entire dataset for each node recorded sequentially, as opposed to each data point being passed through the entire network during each time step, i.e., node 1 receives the entire dataset, then node 2 receives the output of node 1. Furthermore, a 48 node PNN is created with three unique physical systems, where each physical systems state is reset once the data has been passed through. In a future device, each data point must travel

through the PNN in real-time, requiring 48 unique physical systems and measurement set-ups. Therefore, as PNN size increases, so does device complexity. Additionally, outputs from all nodes in the network are recorded and used during the learning phase which, in practice, requires surplus measurement time and memory. Our technique has been tested on systems which take 1D input. For physical systems with >1D inputs, we expect the technique to be equally, if not more, powerful.

Realising a device level PNN requires a reduction in the number of nodes and the number of measured outputs without sacrificing computational performance. This can be achieved by increasing the number of unique physical reservoirs thereby increasing the range of beneficial reservoir dynamics that each node has. Furthermore, the number of measured outputs can be optimised to only record significantly uncorrelated outputs for each node. Optimisation of PNN architecture, and inter-node connectivity (i.e., combining multiple output channels as input to a particular node) will allow further improvements.

We now assess the scalability of the nanomagnetic array-based computing scheme presented in this work. The computing architecture can be separated into three components: input, readout and weight multiplication. At present we use global magnetic field inputs which are unsuitable for scaling due to the large power, long rise-time and large spatial footprint. Ferromagnetic resonance-based readout, whilst theoretically fast, is currently achieved with a laboratory scale RF-source and a lock-in amplifier, resulting in slow data throughout (~20 s for one spectra) and high energy cost (Supplementary Note 13 calculates the power, energy and time for our current scheme compared to conventional processors). To improve the technological prospects of our scheme, input must be delivered on-chip and readout powers and speeds must be reduced to be competitive with alternative approaches.

We now propose and benchmark potential ways to translate nanomagnetic arrays to device level (extended analysis and discussion is provided in Supplementary Note 13). We assume a nanomagnetic array with dimensions of $5\,\mu m \times 5\,\mu m$ (~100 elements). By patterning the array on top of a current-carrying microstrip, array-specific Oersted fields can be generated. For a $5\,\mu m \times 5\,\mu m$ array patterned on top of an Cu microstrip, with width $w = 5\,\mu m$ and thickness $t = 50\,nm$ in height. The current required to produce a 25 mT Oersted field is ~200 mA[47] (see Supplementary Note 13 for calculation details). For Cu resistivity of $16.8\,n\Omega\,m$[48], the resistance is $R = 0.336\,\Omega$ consuming a minimum power $P = 13.4\,mW$. For a pulse time of 1 ns, this equates to an energy of 13.4 pJ per input. Nanomagnet coercivity and therefore power consumption can be reduced by fabricating thinner nanomagnets.

Instead, we can fabricate arrays in contact with a high spin-Hall angle material to switch macrospins via spin-orbit torque[49]. This can be achieved with Ta thickness and magnetic layer thicknesses of around 5 nm and 1–2 nm respectively[49] and current densities on the order of $10^{11}\,Am^{-2}$. Whilst reducing thickness reduces dipolar coupling, arrays of nanomagnets remain highly correlated at these thicknesses[50,51] and will retain the collective processing that thicker nanomagnets exhibit. For a $5\,\mu m \times 5\,\mu m$ Ta strip with 5 nm thickness, the corresponds to a current of 2.5 mA. For Ta resistivity of $131\,n\Omega m$[52], $R_{Ta} = 26.2\,\Omega$ and P = 0.164 mW. Here, switching times can be as low as 250 ps[53] giving input energies of 41 fJ. Spin-orbit torques decrease linearly with increasing ferromagnet thickness[54]. For 20 nm thick nanomagnets used in this work, the current density required to switch an element will be on the order of $10-40\times$ (based on micromagnetic simulations) due to thicker elements and higher coercive fields, giving currents in the range of 25–100 mA and powers in the range of 16.8–269 mW. Limited spin-diffusion in ferromagnets may further increase or even prevent switching in thicker

elements. As such, for thicker nanomagnets, Oersted field switching may be preferable.

Microwave readout can be implemented either sequentially (frequency-swept RF-source) or in parallel (multiple channels simultaneously) depending on the technique. For sequential readout out (as implemented in this work), we can utilise spin-torque FMR which converts magnetisation resonance to DC voltage. Assuming microwave powers in the range of 1–10 mW, the generated voltage at resonance is on the order 1–10 μV which can be amplified with a CMOS low noise high gain amplifier. Readout time per channel is theoretically limited to the time taken to reach steady precession ( ~ 14 ns[55]). As the generated voltage is DC, spectral information is lost, hence each channel must be recorded sequentially, giving a measurement time of ~ 2.6 μs and energy of 2.8 nJ (Supplementary Note 13), dominating over the input speeds. Alternatively, we can pass mixed frequency RF-signals through a device patterned on a co-planar waveguide and detect the microwave absorption, for example via an RF noise source. Here, the readout power is higher (150 mW) but speed and energy are reduced (14 ns and 2.1 nJ respectively, Supplementary Note 13). The output signal can, for example, be sent to multiple spin-torque oscillators tuned to different frequencies serving as RF-diodes[55,56] allowing parallel detection of multiple channels at  ~ ns timescales. Weight multiplication can be achieved by passing output signals to memristor cross-bar arrays which routinely operate with nW–μW powers[57]. As such, the power consumption, energy and time required to perform an operation of our proposed device is on the order of 1–160 mW, 1.5–2800 ns, and 2.1–2.9 nJ depending on the input and readout method used. At the lower end, this is within the power range for battery-less operation using energy harvesting technology[58,59]. For PNNs, nodes are measured sequentially hence the total operating power remains unchanged, and time and energy scale linearly with the number of nodes. Simultaneous measurement of physical nodes would increase power and reduce time. Supplementary Note 13 provides further comparison to CMOS hardware by calculating the number of FLOPs to update an echo-state network (the closest software analogue to nanomagnetic hardware). Our calculations indicate that, for large echo-state networks, the proposed devices have potential to operate at lower energy costs that conventional hardware.

To conclude, here we mitigate the limitations of physical reservoir computing by engineering networks of physical reservoirs with distinct properties. We have engineered multiple nanomagnetic reservoirs with varying internal dynamics, evaluating their computational metrics and performance across a broad benchmark taskset. Our results highlight the computational performance gained from enriched state spaces, applicable across a broad range of neuromorphic systems. PW outperforms MS by up to 31.6× , demonstrating careful design of system geometry and dynamics is critical, with computational benefits available via physical system design optimisation.

We then constructed a physical neural network from a suite of distinct physical systems where interconnections between reservoirs are made virtually and outputs are combined offline, overcoming the fundamental limitation of the memory/NL tradeoff that roadblocks neuromorphic progress. We demonstrate that methods used to improve computational performance in software echo-state networks can be transferred to physical reservoir computing systems. The modular, reconfigurable physical neural network architecture enables strong performance at a broader range of tasks and allows the implementation of modern machine learning approaches such as meta-learning. The high dimensionality enabled by the physical neural network architecture allows us to demonstrate the benefits of operating physical neuromorphic computing in an overparameterised regime to accomplish few-shot learning tasks with just a handful of distinct physical systems. Currently, network interconnections are made virtually. We expect the same performance improvements to be achieved with the next-generation of our scheme with real-time

physical interconnections. Networks of complex nodes described here are largely unexplored, as such many open questions remain. Exploration of different network architectures from both a computational and device architecture perspective is crucial for optimising performance and for device fabrication where increasing the number of nodes and interconnections comes at a cost.

Our method of interconnecting network layers via assessing output-channel/feature metrics allows tailoring of the network metrics, bypassing costly iterative approaches. The approach is broadly-applicable across physical neuromorphic schemes. If the required MC and NL for a given task are known, metric programming allows rapid configuration of an appropriate network. If the required MC and NL are not known or the task depends on more than these metrics, metric programming can be used to search the MC, NL phase space for pockets of high performance. The introduction of a trainable inter-layer parameter opens vast possibilities in implementing hardware neural networks with reservoirs serving as nodes and inter-layer connections serving as weights[60].

## Methods

The Methods section is organised as follows: experimental methods includes the fabrication of samples, measurement of FMR response and the details of implementing reservoir computing and interconnecting arrays. Following this we discuss the tasks chosen and how they are evaluated in Task selection. Learning algorithms then provides a detailed description of the learning algorithms used in this work.

### Experimental methods

**Nanofabrication.** Artificial spin reservoirs are fabricated via the electron-beam lithography liftoff method on a Raith eLine system with PMMA resist. 25 nm $Ni_{81}Fe_{19}$ (permalloy) is thermally evaporated and capped with 5 nm $Al_2O_3$. For WM, a staircase subset of bars are increased in width to reduce its coercive field relative to the thin subset, allowing independent subset reversal via global field. For PW, a variation in widths are fabricated across the sample by varying the electron-beam lithography dose. Within a 100 μm × 100 μm write-field, the bar dimensions remain constant. The flip-chip FMR measurements require mm-scale nanostructure arrays. Each sample has dimensions of roughly ~3 × 2 mm². As such, the distribution of nanofabrication imperfections termed quenched disorder is of greater magnitude here than typically observed in studies on smaller artificial spin systems, typically employing 10–100 micron-scale arrays. The chief consequence of this is that the Gaussian spread of coercive fields is over a few mT for each bar subset. Smaller artificial spin reservoir arrays have narrower coercive field distributions, with the only consequence being that optimal applied field ranges for reservoir computation input will be scaled across a corresponding narrower field range, not an issue for typical 0.1 mT or better field resolution of modern magnet systems.

**Magnetic force microscopy measurement.** Magnetic force micrographs are produced on a Dimension 3100 using commercially available normal-moment MFM tips.

**Ferromagnetic resonance measurement.** Ferromagnetic resonance spectra are measured using a NanOsc Instruments cryoFMR in a Quantum Design Physical Properties Measurement System. Broadband FMR measurements are carried out on large area samples (~3 × 2 mm²) mounted flip-chip style on a coplanar waveguide. The waveguide is connected to a microwave generator, coupling RF magnetic fields to the sample. The output from waveguide is rectified using an RF-diode detector. Measurements are done in fixed in-plane field while the RF frequency is swept in 20 MHz steps. The DC field are then modulated at 490 Hz with a 0.48 mT RMS field and the diode voltage response measured via lock-in. The experimental spectra show the derivative output of the microwave signal as a function of field and frequency. The normalised differential spectra are displayed as false-colour images with symmetric log colour scale.

**Data input and readout.** Reservoir computing schemes consist of three layers: an input layer, a hidden reservoir layer and an output layer corresponding to globally applied fields, the nanomagnetic reservoir and the FMR response, respectively. For all tasks, the inputs are linearly mapped to a field range spanning 35–42 mT for MS, 18–23.5 mT for WM and 30–50 mT for PW, with the mapped field value corresponding to the maximum field of a minor loop applied to the system. In other words, for a single data point, we apply a field at $+\mathbf{H}_{app}$ then $-\mathbf{H}_{app}$. After each minor loop, the FMR response is measured at the applied field $-\mathbf{H}_{app}$ between 8 and 12.5 GHz, 5 and 10.5 GHz, and 5–10.5 GHz in 20 MHz steps for MS, WM, and PW, respectively. The FMR output is smoothed in frequency by applying a low-pass filter to reduce noise. Eliminating noise improves computational performance[5]. For each input data-point of the external signal $s(t)$, we measure ≈300 distinct frequency channels and take each channel as an output. This process is repeated for the entire dataset with training and prediction performed offline.

**Interconnecting arrays.** When interconnecting arrays, we first input the original Mackey-Glass or sinusoidal input into the first array via the input and readout method previously described. We then analyse the memory and NL of each individual frequency output channel (described later). A particular frequency channel of interest is converted to an appropriate field range. The resulting field sequence is then applied to the next array via the computing scheme previously described. This process is then repeated for the next array in the network. The outputs from every network layer are concatenated for learning.

## Task selection

Throughout this manuscript, we focus on temporally driven regression tasks that require memory and NL. Considering a sequence of T inputs $[\mathbf{s}(1), \mathbf{s}(2), \ldots, \mathbf{s}(T)]$, the physical system response is a series of observations $[\mathbf{o}(1), \mathbf{o}(2), \ldots, \mathbf{o}(T)]$ across time. These observations can be gathered from a single reservoir configuration, as in Fig. 1, or can be a collection of activities from multiple reservoirs, in parallel or interconnected, as in Fig. 2. In other words, the response of the system $\mathbf{o}(t)$ at time $t$ is the concatenation of the outputs of the different reservoirs used in the architecture considered. The tasks faced can be divided into five categories:

**Sine transformation tasks.** The system is driven by a sinusoidal periodic input $s(t) = \sin(t)$ and asked to predict different transformations, such as $\bar{\mathbf{y}}(t) = (|\sin(t/2)|, \sin(2t), \sin(3t), \sin^2(t), \sin^3(t), \cos(t), \cos(2t), \cos(3t), saw(t), saw(2t), \ldots)$. The inputs $[s(t), s(t + \delta t), \ldots]$ are chosen to have 30 data points per period of the sinusoidal wave, thus with $\delta t = 2\pi/30$. The total dataset size is 250 data points. If the target is symmetric with respect to the input, the task only requires NL. If the target is asymmetric, then both NL and memory are required.

**Mackey-glass forecasting.** The Mackey-Glass time-delay differential equation takes the form $\frac{ds}{dt} = \beta \frac{s_\tau}{1 + s_\tau^n} - \lambda s$ and is evaluated numerically with $\beta = 0.2$, $n = 10$ and $\tau = 17$. Given $s(t)$ as external varying input, the desired outputs are $\bar{\mathbf{y}}(t) = (s(t + \delta t), s(t + 2\delta t), \ldots, s(t + M\delta t))$, corresponding to the future of the driving signal at M different times. We use 22 data points per period of the external signal for a total of 250 data points. This task predominantly requires memory as constructing future steps requires knowledge of the previous behaviour of the input signal.

**Non-linear auto-regressive moving average tasks.** Non-linear auto-regressive moving average (NARMA) is a typical benchmark used by the reservoir computing community. The definition of the x-th desired

output is $\bar{y}_x(t) = \text{NARMA}\{s(t')|x\} = As(t'-1) + Bs(t'-1)\sum_{n=1}^x s(t'-n) + Cs(t'-1)s(t'-x) + D$, where the constants are set to $A = 0.3$, $B = 0.01$, $C = 2$, $D = 0.1$. The input signal $s(t')$ is the Mackey-Glass signal, where the variable $t'$ is introduced to account for a possible temporal shift of the input. For $t' = t$, $\bar{y}_x(t)$ is the application of NARMA on the Mackey-Glass signal at the current time $t$, while for $t' = t + 10\delta t$, $\bar{y}_x(t)$ is the result of NARMA on the input signal delayed by ten time steps in the future. The index $x$ can instead vary between one, defining a task with a single temporal dependency, and fifteen, for a problem that requires memory of fifteen inputs. Varying $x$ and $t'$, we can define a rich variety of tasks with different computational complexity.

**Evaluations of MC and NL.** Memory capacity and NL are metrics frequently used for the characterisation of the properties of a physical device. While these metrics do not constitute tasks in the common terminology, we include them in this section for simplicity of explanation. Indeed, we use the same training methodology to measure MC and NL as in the other tasks faced. We evaluate these metrics with the Mackey-Glass time-delay differential equation as input. This gives results that are correlated to conventional MC and NL scores, with some small convolution of the input signal−negligible for our purposes of relatively assessing artificial spin reservoirs and designing network interconnections.

For MC the desired outputs are $\bar{y}(t) = (s(t-\delta t), s(t-2\delta t), \ldots, s(t-k\delta t))$, corresponding to the previous inputs of the driving signal at different times. To avoid effects from the periodicity of the input signal, we set $k = 8$. The $R^2$ value of the predicted and target values is evaluated for each value of $\delta t$ where a high $R^2$ value means a good linear fit and high memory and a low $R^2$ value means a poor fit a low memory. The final MC value is the sum of $R^2$ across all output elements.

For NL, the weights are optimised to map the delayed inputs $(s(t), s(t-\delta t), \ldots, s(t-k\delta t))$ to the device response $\mathbf{o}(t)$, where k = 7. For each output, the $R^2$ value of the predicted and target values is evaluated. NL for a single output is given by $1-R^2$, i.e., a good linear fit gives a high $R^2$ and low NL and a bad linear fit gives low $R^2$ and high NL. NL is averaged over all selected features.

Memory capacity and NL can be calculated using a single multiple frequency channels. For the single-channel analysis, we perform the same calculations but using just a single FMR channel.

**Frequency decomposition, a few-shot learning task.** The network is driven by a sinusoidal input $s(t)$ and needs to reconstruct a decomposition of a temporal varying signal in the form of $\bar{y}(t) = \sum_{n}^{N_\omega} a_n \sin(nt + \theta_n)$, where $N_\omega = 5$. The values of $a_n$ and $\theta_n$ are randomly sampled at the beginning of each task from uniform distributions. In particular, $a_n \in [-1.2\ 1.2]$ and $\theta_n \in [0\ \pi/n]$. The output layer is composed of $N_\omega$ nodes, and the system is asked to predict a target $\bar{y}(t) = (a_1\sin(t + \theta_1), a_2\sin(2t + \theta_2), \ldots, a_5\sin(5t + \theta_5))$ after observing the values of $\bar{y}(t)$ over K data points, i.e., time steps. The value of K adopted for the examples of Fig. 4b, c is fifteen, but the network reports good performance even with $K = 10$ (Fig. 4d, e). To face this challenging task, we use the PNN in the overparameterised regime and a meta-learning algorithm to quickly adapt the readout connectivity. Details of the meta-learning approach are given below in the Meta-learning section of the Methods.

**Learning algorithms**
The type of learning algorithm we use to select features and train the networks varies throughout the manuscript. For comparisons between different systems (Fig. 12), training is accomplished through a features selection algorithm (discussed below) and optimisation of the readout weights $\mathbf{W}_o$. When exploring the effects of overparameterisation, we randomly select features and then optimise $\mathbf{W}_o$ for those features. We call $\mathbf{x}(t)$ the representation at which the readout weights operate, and

we define the output of the system as $\mathbf{y}(t) = \mathbf{W}_o\mathbf{x}(t)$. The vector $\mathbf{x}(t)$ simply contains a subset of features of $\mathbf{o}(t)$ defined via the feature selection algorithm or randomly picked as in the dimensional study on overparameterisation. In this setting, optimisation of $\mathbf{W}_o$ is achieved with linear or ridge-regression, which minimises the error function $E = \sum_t ||\bar{y}(t) - \mathbf{W}_o\mathbf{x}(t)||^2 + \lambda||\mathbf{W}_o||^2$.

For the simulations where we show the adaptability of the system with a limited amount of data (results of Fig. 4 and of Section Learning in the overparameterised regime), we used gradient-descent optimisation techniques, particularly Adam[61], to minimise the mean-squared error between prediction and target. All codes and data for the learning algorithms are available online (see code availability statement for details).

**Feature selection.** The dimensionality of an observation $o(t)$ can vary depending on the architecture considered, spanning from $\approx 250$ dimensions when using a single arrays to $\approx 14,000$ for the PNN of Fig. 2. The high readout dimensionality allows better separability of input data, however, high-dimensional spaces constitute a challenge due to overfitting issues. As such, learning over a high-dimensional features' space with few data points constitutes a challenge and opportunity for physically defined reservoirs. For this reason, we design a feature-selection methodology to avoid overfitting and to exploit the computational abilities of architectures with varying complexity (Fig. 5, Algorithms 1, 2). The methodology adopted can be at first described as a 10 cross-validation (inner validation loop) of a 10 cross-validation approach (outer validation loop), where the outer cross-validation is used to accurately evaluate the performance and the inner loop is used to perform feature-selection (Fig. 6). For each split, feature selection is accomplished by discarding highly correlated features and through an evolutionary algorithm. The independent parts of this methodology are known, but the overall procedure is unique and can give accurate performance measurements for our situation, where we have a suite of systems with varying dimensionality to compare over limited data.

---

**Algorithm 1**. Hyperparameter selection

**for** each split in outer loop $i$ **do**
 **for** each split in inner loop $j$ **do**
 **for** each $\theta \in \Theta, \lambda \in \Lambda$ **do**
 Find optimal weights: $\mathbf{W}_o^* = \arg\min_{\mathbf{W}_o} E\{\text{Tr}_{ij}|\mathbf{W}_o, \theta, \lambda\}$
 Compute error on the test set $E\{\mathcal{T}'_{ij}|\mathbf{W}_o^*, \theta, \lambda\}$
 **end for**
 Select optimal hyperparameters as $\theta_i^*, \lambda_i^* = \arg\min_{\theta \in \Theta, \lambda \in \Lambda} \sum_{j=1}^{10} E\{\mathcal{T}'_{ij}|\mathbf{W}_o^*, \theta, \lambda\}$
 **end for**
 Find the corresponding boolean vector $\theta_i^* \to \mathbf{f}^{(i)}$
**end for**

---

**Algorithm 2**. Evolutionary Algorithm

**for** for each split in outer loop $i$ **do**
 Initialise parents $\mathbf{F}_p = \{\mathbf{f}(n)\} = \mathbf{f}^{(i)}$
 **repeat**
 Apply crossover and mutations to generate children $\mathbf{F}_c$
 **for** each split in inner loop j **do**
 **for** each $\mathbf{f} \in \mathbf{F}_c$ **do**
 Find optimal weights: $\mathbf{W}_o^* = \arg\min_{\mathbf{W}_o} E\{\text{Tr}_{ij}|\mathbf{W}_o, f, \lambda_i^*\}$
 **end for**
 **end for**
 Select new $\mathbf{F}_p$ as the $\mathbf{f}'$ with minimal $\sum_{j=1}^{10} E\{\mathcal{T}'_{ij}|\mathbf{W}_o^*, \mathbf{f}, \lambda_i^*\}$
 **until** best performance on $\mathcal{V}_i$
**end for**

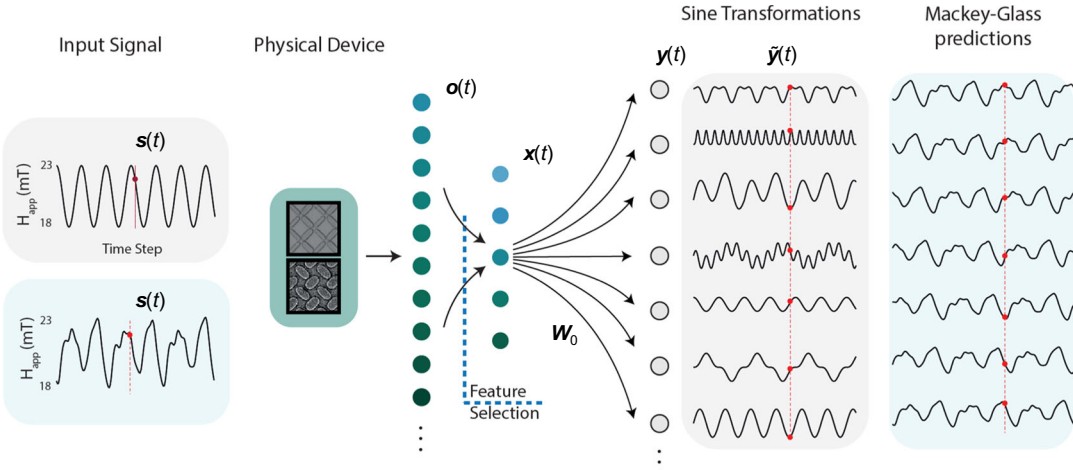

**Fig. 5 | Schematic of offline learning.** Input data is applied to the physical system producing a set of **o**(t) features. Feature selection reduces the size of **o**(t) to a new set of **x**(t) features. We then apply ridge regression to obtain a set of weights **W**ₒ to obtain the transformation/prediction **y**(t) = **W**ₒ**x**(t).

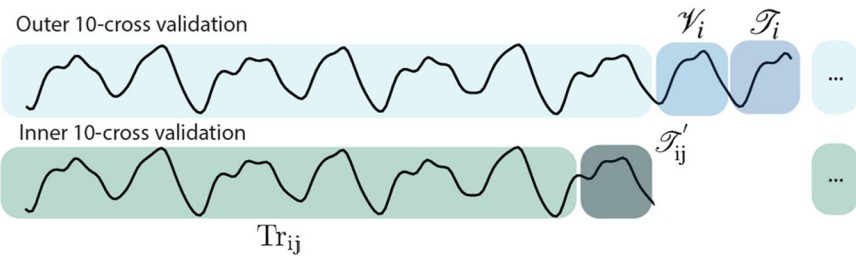

**Fig. 6 | Data splitting during training.** Schematic of the data splits for the inner and outer validation loops.

We now describe the feature selection methodology in detail. Considering the response of a system across time $\mathbf{o}(t)$, the feature selection algorithm aims to select a subset of features $\mathbf{x}(t)$ that will be used for training and evaluation. We can consider feature selection as a boolean operation over the $\mathbf{o}(t)$ feature space, where a value of one (zero) corresponds to the considered feature being used (neglected). If $m$ is the dimensionality of $\mathbf{o}(t)$, the number of possible ways to define $\mathbf{x}(t)$ is $2^m-1$. As a consequence, the feature selection algorithm can also lead to overfitting. Therefore, we must implement an additional cross-validation step to ensure the performance of the selected feature is general across the entire dataset.

Considering a specific split of the outer validation loop (Fig. 6), where we select a validation set $\mathcal{V}_i$ and a test set $\mathcal{T}_i$ (comprising of the 10% of the data each, i.e., 25 data points), we perform 10 cross-validations on the remaining data to optimise hyperparameter values through grid-search. In this inner validation loop, each split corresponds to a test set $\mathcal{T}'_{ij}$ (comprising again of the 10% of the remaining data, without $\mathcal{V}_i$ and $\mathcal{T}_i$), where changing $j$ means to select a different test-split in the inner loop based on the $i$-th original split of the outer validation (Fig. 6). The remaining data, highlighted in green in Fig. 6, are used for training to optimise the readout weights and minimise the error function $E$ through ridge-regression as previously described.

At this stage, we perform a grid-search methodology on hyperparameters $\theta$ and $\lambda$ which control directly and indirectly the number of features being adopted for training (Algorithm 1). The hyperparameter $\theta$ acts as a threshold on the correlation matrix of the features. Simply, if the correlation among two features exceeds the specific value of $\theta$ considered, one of these two features is removed for training (and testing). The idea behind this method is to discard features that are highly correlated, since they would contribute in a

similar way to the output. This emphasises diversity in the reservoir response. The hyperparameter $\lambda$ is the penalty term in ridge-regression. Higher values of $\lambda$ lead to a stronger penalisation on the magnitude of the readout weights. As such, $\lambda$ can help prevent overfitting and controls indirectly the number of features being adopted. We should use a high value of $\lambda$ if the model is more prone to overfitting the training dataset, a case that occurs when the number of features adopted is high. Calling $\mathrm{E}\left\{\mathcal{T}'_{ij}|\mathbf{W}^*_o,\theta,\lambda\right\}$ the error computed on the test set $\mathcal{T}'_{ij}$ with weights $\mathbf{W}^*_o$ optimised on the corresponding training data ($\mathbf{W}^*_o = \arg\min_{\mathbf{W}_o}\mathrm{E}\left\{\mathrm{Tr}_{ij}|\mathbf{W}_o,\theta,\lambda\right\}$ in Algorithm 1 and with hyperparameter values $\theta$ and $\lambda$, respectively, we select the values of the hyperparameters that correspond to the minimum average error over the test sets in the inner validation loop. Otherwise stated, we select the optimal $\theta^*_i$ and $\lambda^*_i$ for the $i$-th split in the outer loop from the test average error in the inner 10 cross-validation as $\theta^*_i,\lambda^*_i = \arg\min_{\theta\in\Theta,\ \lambda\in\Lambda}\sum_{j=1}^{10}\mathrm{E}\left\{\mathcal{T}'_{ij}|\mathbf{W}^*_o,\theta,\lambda\right\}$. This methodology permits to find $\theta^*_i$ and $\lambda^*_i$ that are not strongly dependent on the split considered, while maintaining the parts of the dataset $\mathcal{V}_i$ and $\mathcal{T}_i$ unused during training and hyperparameter selection. The sets $\Theta$ and $\Lambda$ correspond to the values explored in the grid-search. In our case, $\Theta = \{1,\ 0.999,\ 0.99,\ 0.98,\ 0.97,\ ...\}$ and $\Lambda = \{1e-4, 1e-3, 1e-2, 5e-2, 1e-1\}$. Repeating this procedure for each split of the outer loop, we found the optimal $\theta^*_i$ and $\lambda^*_i$, for $i = 1, ..., 10$. This concludes the hyperparameter selection algorithm described in Algorithm 1. Selection of the hyperparameters $\theta^*_i$ permit to find subsets of features based on correlation measures. However, promoting diversity of reservoir measures does not necessarily correspond to the highest performance achievable. Thus, we adopted an

evolutionary algorithm to better explore the space of possible combination of measurements (Algorithm 2).

It is necessary now to notice how a value of $\theta_i^*$ corresponds to a $m$-dimensional boolean vector $\mathbf{f}^{(i)}$, whose $j$-th dimension is zero if its $j$-th feature $f_j^{(i)}$ is correlated more than $\theta_i^*$ with at least one other output. For each split $i$ in the outer loop, we adopted an evolutionary algorithm that operates over the $m$-dimensional boolean space of feature-selection, where each individual corresponds to a specific vector $\mathbf{f}$. At each evolutionary step, we perform operations of crossover and mutation over a set of $N_p$ parents $\mathbf{F}_p = \{\mathbf{f}(n)\}_{n=1,\dots,N_p}$. For each split $i$ of the outer loop and at the first evolutionary step, we initialised the parents of the algorithm to $\mathbf{f}^{(i)}$. We defined a crossover operation among two individuals $\mathbf{f}(i)$ and $\mathbf{f}(j)$ as $\mathbf{f} = \text{CrossOver}(\mathbf{f}(i), \mathbf{f}(j))$ where the $k$-th dimension of the new vector $\mathbf{f}$ is randomly equal to $f_k^{(i)}$ or $f_k^{(j)}$ with the same probability. A mutation operation of a specific $\mathbf{f}(i)$ is defined as $\mathbf{f} = \text{Mutation}(\mathbf{f}(i))$ by simply applying the operator *not* to each dimension of $\mathbf{f}(i)$ with a predefined probability $p_m$. The application of crossovers and mutations permits the definition of a set of $N_c$ children $\mathbf{F}_c = \{\mathbf{f}(n)\}_{n=1,\dots,N_c}$ from which we select the $N_p$ models with the highest performance over the test sets of the inner loop as parents for the next iteration. Otherwise stated, we selected the $\mathbf{F}_p$ vectors corresponding to the lowest values of the average error as $\mathbf{F}_p = \arg\min_{\mathbf{f} \in \mathbf{F}_c}^{N_p} \sum_{j=1}^{10} \mathrm{E}\{\mathcal{T}'_{ij} | \mathbf{W}_o^*, \mathbf{f}, \lambda_i^*\}$ where $\arg\min^{N_p}$ selects the $N_p$ arguments of the corresponding function with minimal values. We notice how a step of the evolutionary approach aims to minimise an error estimated in the same fashion as in the algorithm of Algorithm 1, but this time searching for the best performing set $\mathbf{F}_p$, rather then the best performing couple of hyperparameter values $\lambda_i^*$ and $\theta_i^*$.

Finally, we stop the evolutionary algorithm at the iteration instance where the average performance of $\mathbf{F}_p$ over $\mathcal{V}_i$ is at minimum and selected the model $\mathbf{f}_i^*$ with the lowest error on $\mathcal{V}_i$. The utilisation of a separate set $\mathcal{V}_i$ for the stop-learning condition is necessary to avoid overfitting of the training data. Indeed, it is possible to notice how the performance on $\mathcal{V}_i$ would improve for the first iterations of the evolutionary algorithm and then become worse. This concludes the evolutionary algorithm in Algorithm 2. At last, the overall performance of the model is computed as the sum of the mean-squared errors over the outer validation loop as $\mathcal{E} = \sum_{i=1}^{10} \mathrm{E}\{\mathcal{T}_i | \mathbf{W}_o^*, \mathbf{f}_i^*, \lambda_i^*\}$. Summarising, we can think the overall methodology as an optimisation of relevant hyperparameters followed by a fine-tuning of the set of features used through an evolutionary algorithm. The final performance and its measure of variation reported in the paper are computed as average and standard deviation over ten repetitions of the evolutionary algorithm, respectively.

**Evaluating overparameterisation.** To explore the effects of overparameterisation it is necessary to vary the number of network parameters (i.e., number of FMR channels). This is achieved as follows: first, a set of tasks is prepared for which the MSE will be evaluated. Here, we analyse performance when predicting future values of the Mackey-Glass equation. We average MSE when predicting $t+1$, $t+3$, $t+5$, $t+7$, $t+9$ and $t+11$. Next, we vary the number of parameters. To do so, we randomly shuffle a sequence of integers from 0 to $N$, where $N$ is the total number of output channels for a given network e.g., $N = (328, 34, 273\dots)$. From this, we select the first $P$ points of the sequence, giving a list of indexes referring to which channels to include in that sequence. For those channels, we perform a 5 cross-validation of different train and test splits and use linear regression to evaluate the train and test MSE for a given task and set of outputs. $P$ is increased in steps of $N/500$ for single, parallel, and series networks and steps of $N/5000$ for the PNN. For a given task, we repeat this process for 50 random shuffles to ensure we are sampling a

broad range of output combinations for each network. The displayed MSEs are the average and standard error of these 50 trials over all tasks.

## Meta-learning

**Algorithm 3.** Meta-learning through MAML

> **Sample** a batch of tasks from $p(\mathcal{T})$
> **for** each task $i$ **do**
> > **Sample** K datapoints of device responses $\mathcal{D}_i = \{\dots, (\bar{\mathbf{y}}(t_k), \mathbf{o}(t_\mathbf{k})), \dots\}$, $k = 1, \dots, K$
> > **for** number of inner loop steps $n = 1, \dots, \tilde{n}$ **do**
> > > $\mathbf{y}(t_k) = f(\mathbf{o}(t_k) | \boldsymbol{\alpha}_i(n), \mathbf{W}_o)$
> > > $\boldsymbol{\alpha}_i(n+1) = \boldsymbol{\alpha}_i(n) - \eta_1 \nabla_{\boldsymbol{\alpha}_{i(n)}} \mathrm{E}_{\mathcal{T}_i}(\mathcal{D}_i | \boldsymbol{\alpha}_i(n), \mathbf{W}_o)$
> > > **Sample** Q datapoints of devices response $\mathcal{D}'_i = \{\dots, (\bar{\mathbf{y}}(t_q), \mathbf{o}(t_\mathbf{q})), \dots\}$, $q = 1, \dots, Q$ for the meta-update
> > **end for**
> > **Perform** the meta-update
> > $\mathbf{W}_o \leftarrow \mathbf{W}_o - \eta_2 \nabla_{\mathbf{W}_o} \sum_i \mathrm{E}_{\mathcal{T}_i}(\mathcal{D}'_i | \boldsymbol{\alpha}_i(\tilde{n}), \mathbf{W}_o)$
> > $\boldsymbol{\alpha}(0) \leftarrow \boldsymbol{\alpha}(0) - \eta_2 \nabla_{\boldsymbol{\alpha}(0)} \sum_i \mathrm{E}_{\mathcal{T}_i}(\mathcal{D}'_i | \boldsymbol{\alpha}_i(\tilde{n}), \mathbf{W}_o)$
> **end for**

The goal of meta-learning is optimise an initial state for the network such that when a new task with limited data points is presented, the network can be quickly updated to give strong performance. A schematic of the meta-learning algorithm is presented in Fig. 7 and pseudo-code shown in Algorithm 3.

Let us consider a family of M tasks $\mathcal{T} = \{\mathcal{T}_1, \mathcal{T}_2, \dots, \mathcal{T}_M\}$. Each task is composed of a dataset and a cost function $\mathrm{E}_{\mathcal{T}_i}$. A meta-learning algorithm is trained on a subset of $\mathcal{T}$ and asked to quickly adapt and generalise on a new test subset of $\mathcal{T}$. Otherwise stated, the aim of meta-learning is to find an initial set of parameters $\mathbf{W}(0)$ that permits learning of an unseen task $\mathcal{T}_i$ by updating the model over a small number of data points from $\mathcal{T}_i$. To achieve this, we use a variation of the MAML algorithm, which is now summarised. For a given task $\mathcal{T}_i$, the initial set of trainable parameters $\mathbf{W}(0)$ are updated via gradient descent over a batch of data points $\mathcal{D}_i = \{\dots, (\mathbf{o}_j, \bar{\mathbf{y}}_j), \dots\}$, where $\mathbf{o}_j$ and $\bar{\mathbf{y}}_j$ are the inputs and targets for the j-th data point respectively

$$\mathbf{W}_i(n) = \mathbf{W}_i(n-1) - \eta_1 \nabla_{\mathbf{W}_{i(n-1)}} \mathrm{E}_{\mathcal{T}_i}(\mathcal{D}_i | \mathbf{W}_i(n-1)) \quad (1)$$

where $\eta_1$ is the learning rate adopted. Eq.(1) is repeated iteratively for $n = 0, \dots, \tilde{n}$, where $\tilde{n}$ are the number of updates performed in each task. We notice how the subscripts $i$ on the parameters $\mathbf{W}$ are introduced because the latter become task-specific after updating, while they all start from the same values $\mathbf{W}(0)$ at the beginning of a task. MAML optimises the parameters $\mathbf{W}(0)$ (i.e., the parameters used at the start of a task, before any gradient descent) through the minimisation of cost functions sampled from $\mathcal{T}$ and computed over the updated parameters $\mathbf{W}(\tilde{n})$ i.e., the performance of a set of $\mathbf{W}(0)$ is evaluated based on the resulting $\mathbf{W}(\tilde{n})$ after gradient descent. Mathematically, the aim is to find the optimal $\mathbf{W}(0)$ that minimises the meta-learning objective $\mathcal{E}$

$$\mathcal{E} = \sum_{\mathcal{T}} \mathrm{E}_{\mathcal{T}_i}(\mathcal{D}'_i | \mathbf{W}_i(\tilde{n})) \quad (2)$$

$$\mathbf{W}(0) \leftarrow \mathbf{W}(0) - \eta_2 \nabla_{\mathbf{W}(0)} \sum_i \mathrm{E}_{\mathcal{T}_i}(\mathcal{D}'_i | \mathbf{W}_i(\tilde{n})) \quad (3)$$

where the apex ' is adopted to differentiate the data points used for the meta-update from the inner loop of Eq. (1), and $\eta_2$ is the learning rate for the meta-update. Gradients of $\mathcal{E}$ need to be computed with respect to $\mathbf{W}(0)$, and this results in the optimisation of the recursive Eq.(1) and the computation of higher-order derivatives. In our case, we use the first-order approximation of the algorithm[31]. After the meta-learning process,

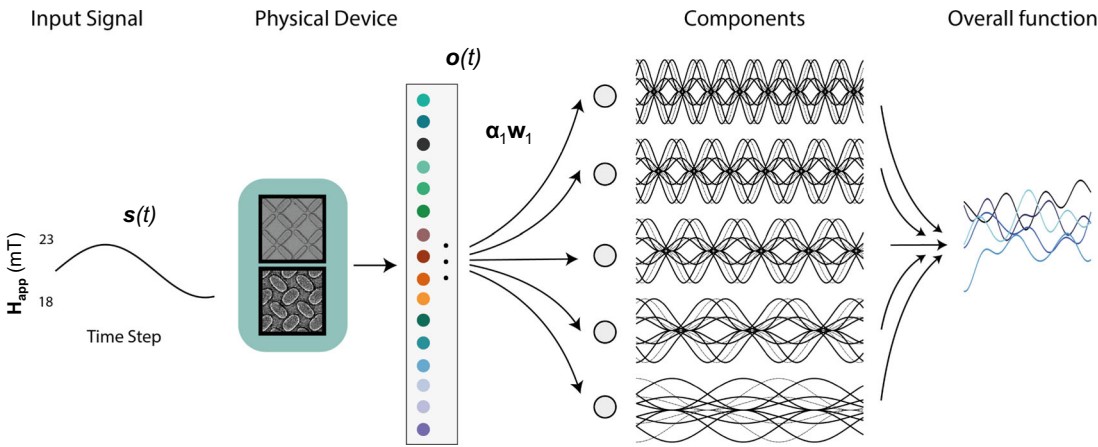

**Fig. 7 | Schematic of the few-shot learning task.** The input signal is applied to the multilayer neural network, and the observations are recorded. The system is then trained to learn a subset of frequency decomposition with minimal training data.

the system is asked to learn an unseen task $\mathcal{T}_j$ updating $\mathbf{W}(0)$ through the iteration of Eq.(1) computed on a small subset of data of $\mathcal{D}_j$.

In contrast to previous works, we adopted this learning framework on the response of a physical system. The optimisation occurs exclusively at the readout level, leaving the computation of non-linear transformations and temporal dependencies to the physical network. The outcome of our application of meta-learning, depends on the richness of the dynamics of the nanomagnetic arrays. In this case, the readout connectivity matrix is not task-specific as in previous sections, we use the same $\mathbf{W}(0)$ for different tasks.

We decompose the parameters of the system $\mathbf{W}$ into two sets of parameters $\{\mathbf{W}_o, \boldsymbol{\alpha}\}$ to exploit few-shot learning on the readout from the PNN. The output of the system is defined as $\mathbf{y}(t) = f(\mathbf{o}(t)|\boldsymbol{\alpha}, \mathbf{W}_o)$, where the function $f$ is linear and follows $f(\mathbf{o}(t)|\boldsymbol{\alpha},\mathbf{W}_o) = \boldsymbol{\alpha} \bigcirc (\mathbf{W}_o \mathbf{o}(t))$ where $\bigcirc$ stands for element by element multiplication. The inclusion of parameters $\boldsymbol{\alpha}$ as scaling factors for the weights has previously been used in[62], albeit in the context of weight normalisation. Here, we train the parameters $\boldsymbol{\alpha}$ and $\mathbf{W}_o$ with different timescales of the learning process, disentangling their contribution in the inner (task-specific updates, Eq.(1)) and outer loops (computation of the meta-learning objective of Eq.(2)) of the MAML algorithm. Specifically, the $\boldsymbol{\alpha}$ parameters are updated for each task following

$$\boldsymbol{\alpha}_i(n+1) = \boldsymbol{\alpha}_i(n) - \eta_1 \boldsymbol{\nabla}_{\boldsymbol{\alpha}_i(n)} \mathrm{E}_{\mathcal{T}_i}\big(\mathcal{D}_i|\boldsymbol{\alpha}_i(n), \mathbf{W}_o\big) \text{ for } n = 1, \dots, \tilde{n} \quad (4)$$

while the parameters $\mathbf{W}_o$ are optimised through the meta-learning objective via

$$\mathbf{W}_o \leftarrow \mathbf{W}_o - \eta_2 \sum_j \mathrm{E}_{\mathcal{T}_j}(\mathcal{D}_j|\boldsymbol{\alpha}_j(\tilde{n})) \quad (5)$$

In this way, training of the parameters $\mathbf{W}_o$ is accomplished after appropriate, task-dependent scaling of the output activities. A pseudocode of the algorithm is reported in Fig. 7.

**Echo state network comparison.** The Echo-state network of Fig. 2g–i is a software model defined through

$$\mathbf{x}(t+\delta t) = (1 - \delta t/\tau)\mathbf{x}(t) + \delta t/\tau\, f\big(\mathbf{W}_{in}\mathbf{s}(t) + \mathbf{W}_{esn}\mathbf{x}(t)\big) \quad (6)$$

where $\mathbf{x}$ is the reservoir state, $\mathbf{s}$ is the input, $\mathbf{W}_{esn}$ and $\mathbf{W}_{in}$ are fixed and random connectivity matrices defined following standard methodologies[63] and $\tau$ is a scaling factor. In particular, the eigenvalues of

the associated, linearised dynamical system are rescaled to be inside the unit circle of the imaginary plane. Training occurs on the readout level of the system. Echo-state networks and their spiking analogous liquid-state machines[64] are the theoretical prototypes of the reservoir computing paradigm. Their performance can thus constitute an informative reference for the physically defined networks. We highlight two differences when making this comparison: first, the ESN is defined in simulations and it is consequently not affected by noise; second, the physically defined network has a feed-forward topology, where the memory of the system lies in the intrinsic dynamics of each complex node and the connectivity is not random but tuned thanks to the designed methodology.

For the results of Fig. 2, we evaluate the performance of a 100-node ESN. For each value of the dimensionality explored, we repeated the optimisation process ten times resampling the random $\mathbf{W}_{in}$ and $\mathbf{W}_{esn}$. The black dots in Fig. 2 report the average performance across these repetitions as the number of nodes varies. The black line reflects a polynomial fit of such results for illustrative purposes, while the grey area reflects the dispersion of the distributions of the results.

**Multilayer perceptron comparison.** The multilayer perceptron (MLP) comparison in the supplementary information is defined as follows. We have an MLP with four layers: an input layer, two hidden layers interconnected with a ReLu activation function and an output layer. We vary the size of the hidden layers from 1 to 500. Standard MLP's do not have any recurrent connections and therefore do not hold information about previous states, hence predictive performance is poor. We add false memory by providing the network with the previous $T_{seq}$ data points as input i.e., $\tilde{\mathbf{s}}(t) = (s(t), s(t-1), \dots, s(t - T_{seq} + 1))$. We vary $T_{seq}$ from 1 to 10 where $T_{seq} = 1$ corresponds to a standard MLP only seeing the current input data point. Learning of MLP weights is performed using gradient descent, specifically Adam[61].

## Data availability

The experimental data used in this study is available at Github and Zenodo[32] under accession codes https://github.com/StenningK/NeuroOverParam.git and https://doi.org/10.5281/zenodo.12721639.

## Code availability

The code developed in this study is available at Github and Zenodo[32] under accession codes https://github.com/StenningK/NeuroOverParam.git and https://doi.org/10.5281/zenodo.12721639.

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

## Acknowledgements

K.D.S. was supported by The Eric and Wendy Schmidt Fellowship Program and the Engineering and Physical Sciences Research Council (Grant no. EP/W524335/1). This work was supported by the EPRSC grant EP/X015661/1 to W.R.B. and H.K. J.C.G. was supported by the Royal Academy of Engineering under the Research Fellowship programme and the EPSRC ECR International Collaboration Grant 'Three-Dimensional Multilayer Nanomagnetic Arrays for Neuromorphic Low-Energy Magnonic Processing' EP/Y003276/1. A.V. was supported by the EPSRC Centre for Doctoral Training in Advanced Characterisation of Materials (Grant no. EP/L015277/1) and EPSRC grant EP/X015661/1. K.E.S. acknowledges funding from the German Research Foundation (Project no. 320163632) and from the Emergent AI Centre funded by the Carl-Zeiss-Stiftung. C.C. and T.C. performed the work as part of their Physics MSci project at Imperial College London. H.H. was supported by an EPSRC Doctoral Prize Fellowship EP/W524323/1. Analysis was performed on the Imperial College London Research Computing Service (https://doi.org/10.14469/hpc/2232). The authors would like to thank David Mack for excellent laboratory management. The authors thank Matthew O.A. Ellis for the insightful and inspiring discussions regarding optimisation.

## Author contributions

K.D.S. and J.C.G. conceived the work and directed the project throughout. K.D.S. drafted the manuscript with contributions from all authors in the editing and revision stages. K.D.S. and L.M. implemented the computation schemes. L.M. developed the cross-validation training approach for reducing overfitting on shorter training datasets. L.M. developed the feature selection methodology for selecting optimal features from many reservoirs in the multilayer physical neural network architecture. L.M. designed and implemented the meta-learning scheme. K.D.S. designed and implemented the method of interconnecting networks. C.C. and T.C. aided in analysis of reservoir metrics. K.D.S., J.C.G., and A.V. performed FMR measurements. J.C.G. and H.H. performed M.F.M. measurements. J.C.G. and K.D.S. fabricated the samples. J.C.G. and A.V. performed CAD design of the structures. J.C.G. performed scanning electron microscopy measurements. J.L. contributed task-agnostic metric analysis code. F.C. helped with conceiving the work, analyzing computing results, and providing critical feedback. E.V. provided oversight on computational architecture design. K.E.S., E.V., and H.K. provided critical feedback. W.R.B. oversaw the project and provided critical feedback.

## Competing interests

Authors patent applicant. Inventors (in no specific order): K.D.S., J.C.G., A.V., H.H., W.R.B. Application number: PCT/GB2022/052501. Application filed. Patent filed in the UK through Imperial College London. The patent covers the method of programming deep networks. The remaining authors declare no competing interests.
