## [Peer Review File · Nature Communications]

Neuromorphic Overparameterisation, Generalisation and Few-Shot Learning in Multilayer Physical Neural NetworksREVIEWER COMMENTS

Reviewer #1 (Remarks to the Author):

This article by Stenning et al. proposes to interconnect physical reservoirs (each implemented by an artificial spin ice array) to perform computing. The main findings of the paper are:

- Different types of Artificial Spin Ice patterns provide different computing functionalities, and therefore how they are chosen and interconnected is key for computing. For instance, connecting a reservoir with short term memory to a reservoir with long term memory provides better accuracy on a time series task than the reverse.
- The total system of interconnected reservoirs can reach the over-parametrized regime. In this regime, overfitting is overcome and parameters increase leads to accuracy increase. Furthermore, the system is then able to learn from much fewer examples than outside of the overparametrized regime.

I think this is interesting, in particular the observation of the over-parametrized regime. However, I think the work needs major revisions before being considered for publication in Nature Communications, as there are many unanswered questions as well as uncertainties regarding the significance of the work.

Concerns about the content:

- The claims of the paper rely on the ability to interconnect reservoirs and perform a large matrix multiplication for the final layer. However, here this is performed fully offline by combining recorded data. I think that this should be made crystal clear upfront in the paper because the claims in the introduction can be misleading in that regard.
- Because this interconnection and matrix multiplication are critical, there needs to be a clear and realistic path towards implementing it in hardware. I fully agree that it makes sense to investigate the computing capability of interconnected reservoirs before launching the difficult hardware implementation. However, I also think that -- in order for this work to be significant -- there needs to be a more rigorous evaluation of how the hardware implementation would work. I detail my questions about it below.
- The input for now is a magnetic field. How would this work in a fully hardware implementation? How does this limit the parallelism, compactness, speed and energy efficiency? Do the authors have in mind alternatives to a field input? If yes, what are these and how would they perform regarding the metrics mentioned above.
- The overparametrized regime is here reached with a few thousand parameters. This requires a matrix multiplication of that dimension. Here it is implemented in software. How do the authors plan this in a real system? For real life tasks, I suppose that this size will even go up. As large matrix multiplication is usually the bottleneck in terms of time and energy consumption for neural networks, doing it in software seems limiting to me.
- The introduction puts forward the claim that the proposed system offers a “high-dimensional data output” while “Neuromorphic systems tend to employ one or < 10 physical output channels”. Memristive crossbar arrays are made with over 500 outputs routinely and

state of the art phase change memory chips have tens of millions of trainable parameters. The most recent and most impressive example is Ambrogio, S., Narayanan, P., Okazaki, A. et al. An analog-AI chip for energy-efficient speech recognition and transcription. *Nature* 620, 768–775 (2023) but there are many others. So, I do not think that this claim is valid.

Furthermore, in the proposed system, the multiple outputs are for now fully realized by offline post treatment. How would multiple outputs work in a hardware implementation? It seems that it would require either sending a single frequency signal at a time (sequential) or one multi-frequency signal (parallel). Which option is considered and how would it be implemented (signal generation, frequency selection etc.)? If the parallel option is considered, are there any studies about the ability of artificial spin ice arrays to react to multiple signals simultaneously?

- Exactly what are the trainable parameters and how they are counted is not very clear. How independent are they? How would this change when in a real hardware system with constraints on the input/outputs. In a fully hardware implementation, how would these parameters be stored? Do the authors envision offline or online training?

- At equal number of trainable parameters, how do the different systems (single reservoir, multiple reservoirs) compare? For instance, Figure 3c compares different reservoir types but the number of parameters and number of reservoirs are not addressed clearly. In Figure 3b, the "PNN" has better accuracy than the other systems, but also has the largest number of nodes. How does the nodes number relate to the parameters number? At equal number of nodes, do the other systems perform as well as the PNN?

- More generally, the proposed system is not rigorously compared to conventional hardware in terms of energy efficiency / size / speed. Thus, it is hard to assess the relevance of the work.

- What sets the number of parameters to reach overparameterization? Does it depend on the task, number of examples, network? For different architectures of reservoir systems, is it reached at the same number?

- Is a specific design of the reservoir system needed for each task? Also, how is a design reproducible for different reservoirs that are nominally identical, i.e. is device variability a problem? How would this design work in applications? Are there possibilities to generalize? How does this work relate to other works on interconnecting reservoirs (such as L. Grigoryeva, J. Henriques, L. Larger and J. Ortega, "Nonlinear Memory Capacity of Parallel Time-Delay Reservoir Computers in the Processing of Multidimensional Signals," in *Neural Computation*, vol. 28, no. 7, pp. 1411-1451, July 2016, doi: 10.1162/NECO_a_00845.)?

Concerns about the presentation:

- The paper is quite dense and honestly it took a lot of work to understand what was actually done. It would really help the readers, especially those not in spintronics or spin ice community, to get a clear description of the system and results and move more technical details to methods and Supplementary.

- Figure 3 is confusing. Too many acronyms are used, that are not legible to outsiders of the subfield. The color code is unclear.

In conclusion, the scientific findings of the study seem sound to me but:

- There are things left to clarify in the over-parametrization study
- Without a better understanding of how the concept would work in a real system and without comparison to conventional hardware, it is difficult to grasp how significant this paper is.
- The claims about output dimension superiority do not seem justified to me.
- The paper's clarity must be improved

Reviewer #2 (Remarks to the Author):

The authors have studied the multilayer physical neural networks (PNNs) based on three types of artificial spin ices (ASIs). The authors quantified the memory capacities and nonlinearities for these three ASIs and benchmarked their performance for Mackey-Glass and sine transformation problems. Then, the authors constructed various types of PNNs based on different combinations of these three ASIs and benchmarked their performance. In the end, they show that the multilayer PNN can realize a double-descent curve and enable few-shot learning. Overall, this study is comprehensive and interesting as it is the first study using non-CMOS technology to show the possibility of realizing an over-parameterized regime. However, I have serious concerns about the benchmark and reproducibility of this work as the key novelty and significance are in the system (algorithm) design, which is highly dependent on the details. If the authors can address these two issues, I am happy to support its publication.

Benchmark issues:

1. Although this paper claims that the PNN with ASIs can realize an over-parameterized regime, it still requires significant resources, including both ASIs and supporting CMOS circuits to process signals between layers. In contrast, CMOS-only multilayer neural networks (software-based) can also realize this over-parameterized regime as refs. 73, 75, and this paper: <https://arxiv.org/abs/2303.14151> show. This suggests the importance of carefully benchmarking the resources to achieve the over-parameterized regime.
 2. While ASIs can realize the over-parameterized regime, nonlinearity is not required at the fundamental level. It has been shown that a double descent curve can happen in ordinary linear regression (<https://arxiv.org/abs/2303.14151>).
- How does a PNN with ASIs perform compared with conventional CMOS systems (software-based) for these simple tasks?
 - What is the key point for the emergence of this double descent curve? Can you use PNN with only one ASI to achieve this double descent curve? If not possible, why? (In my opinion, it should be possible as simple CMOS-only software can achieve it.)
 - If one can use PNN with only one ASI to achieve this double descent curve, what is the performance in the over-parameterized regime?

Reproducibility issues:

1. The authors should show tables to indicate the details of every neural network being used in Fig. 1, Fig. 3, and Fig. 4
 2. I am highly concerned about the reproducibility of this work. The results are highly dependent on the details. Data and codes should be made publicly available to enable reproducibility:
- All figure data for plotted figures
 - Pseudo-code for hardware/software co-design system should be provided in supplementary and the comments of the submitted codes.
 - All codes with raw data for hybrid hardware/software experiments. I understand that intermediate experimental results may not be available. Then, the software codes to collect data and post-process should be shown.

- All codes for purely software simulation to ensure the reproduction of figures like Fig. 1, Fig. 3, and Fig. 4

Reviewer #3 (Remarks to the Author):

The paper proposes using connected artificial spin ice arrays of different geometries for neuromorphic computing. It studies how macrospin square, width-modified square and gradient pinwheel geometries affect the response and explores different ways of connecting these reservoirs into a multi-node network. I find the paper relevant and timely, and the results original and useful to the community. At the same time, I think that it should be more accurate or detailed in several key aspects. If these are properly addressed, the paper can be recommended for publication and will be of interest to the wide readership of Nature Communications.

1. “Neuromorphic schemes typically employ a single physical system, limiting the dimensionality and range of available dynamics – restricting strong performance to a few specific tasks” is a sentence from the abstract that is false. The field of neuromorphic computing is broad and includes some very sophisticated systems (e.g. <https://doi.org/10.1038/s41928-022-00838-3> analog reservoir with output layer; True North architecture from IBM; <https://doi.org/10.1038/s41427-021-00282-3> all-spin networks, etc.). Same applies to the other instances of using “neuromorphic” in the paper, for instance in “tend to employ one or ≤ 10 physical output channels”. Perhaps, replacement with “physical reservoir computing”, which is a small subset of neuromorphic computing, would make it more precise.
2. I struggled to find the boundary between experiments and simulations in this work. If every task has been applied to every architecture experimentally, it should be clarified. If the response of each type of reservoir has been measured once and then fed into a simulation, it should also be explicitly stated.
3. The PNN seems to be the pinnacle of the work, yet its exact structure remains unclear apart from its composition of 52 sub-networks. I could find the information on feature selection in the PNN but not on how and why it has been designed in a certain way.
4. Better understanding of the PNN's complexity and composition is also relevant for comparing its performance to simpler counterparts in Figure S6b. It seems that a significant increase in complexity leads to a rather marginal performance improvement or none (in 50+% cases). Thus, having PW, PW+WM, WM->PW (5 ASI in total) and selecting one depending on the task outperforms PNN (106 ASI) on average. This raises a question: what is the relationship between performance improvement and added network complexity? Would a 15-node PNN yield similar results, or would a 150-node network significantly enhance performance?
5. Continuing question 4, what is the suggested optimal complexity for an individual node? To what extent should these multi-reservoir networks shift toward RNNs to maximize benefits while still maintaining relatively straightforward training? This is particularly important for hardware systems, in which interconnects between reservoirs are “expensive” and complex due to a combination of CMOS and non-CMOS components, such as ASI.
6. The comparison to a software ESN in Figure 3b is ambiguous. The activation function f is undefined. Assuming that it is something typical like \tanh , such a 90-node single-reservoir ESN is fundamentally simpler than a 100+ ASI multi-layer PNN with a) much more complex nodes, b) trainable weights between the nodes. This result can be used as a reference but

not as a direct comparison and not to claim “significantly better performance”.

7. Dimensionality of a physical reservoir and dimensionality of its output are used interchangeably in the discussion of dimensionality in the bottom section of page 1. If a system has 6 physically fabricated readout electrodes, it does not mean that its dimensionality as a reservoir is 6 (it can be anything) but rather that the output dimensionality is 6 (assuming the readouts are not linearly dependent). The FMR-ASI may have a high dimensionality, however, the readout from these dimensions is still done in series (frequency-by-frequency). Therefore, I suggest correcting the statement and answering the question of how the FMR readout dimensionality of the used ASI can be improved.

8. There must be some confusion about MC and NL metrics. For example, MC of PW is 3 on Fig. 2i (lower panel), 5 on Fig. 2j and “4-7” in the text.

Related to this is the question of how MC was determined from the black-blue colormaps in Fig. 2g-i. For example, in 2i) at ~8.7 GHz the memory is completely lost at steps 2-5 (black) but then recovers at step 6-7 (white). How is it possible? And if it is meaningful, could the system recover its memory at step 100 making its MC=100?

9. The authors could add a section discussing the prospects and generalizing their findings including a direction for further improvement, potential problems, what combination of reservoir dynamics could be the most promising, extension to other physical systems etc.

Nitpicking:

1. “Figure 1, point 1” in chapter Network overview is not clear. Perhaps calling it a region would help.

2. More rigorous language might improve the read, counterexamples include “available “for free”, “extremely high dimensionality”, “our efficient method”, all taken from Conclusions section.

3. Should the equation on page 9 be $y(t) = \text{sum}() + y_{\text{OFFSET}}$?

4. What are the colored traces in Fig. 4c?

In conclusion, this is an interesting work delving into the topic of multilayer networks comprised of physical reservoirs. It is impossible to imagine real competition between the proposed system and a software network at this point in time. The same, however, can be said about any physical reservoir. This is why I believe that detailed information on the PNN- and how it was constructed, as well as generalizations applicable to alternative physical reservoirs, could be of significant value to the field. This is also why I think that many practical questions (node connectivity, scalability, power consumption, footprint, etc.) are nonessential and can be omitted. A fair comparison to the software counterpart, in this perspective, is useful as a benchmark but not in the sense of direct competition.

REVIEWER COMMENTS

Reviewer #1 (Remarks to the Author):

This article by Stenning et al. proposes to interconnect physical reservoirs (each implemented by an artificial spin ice array) to perform computing. The main findings of the paper are:

- Different types of Artificial Spin Ice patterns provide different computing functionalities, and therefore how they are chosen and interconnected is key for computing. For instance, connecting a reservoir with short term memory to a reservoir with long term memory provides better accuracy on a time series task than the reverse.

- The total system of interconnected reservoirs can reach the over-parametrized regime. In this regime, overfitting is overcome and parameters increase leads to accuracy increase. Furthermore, the system is then able to learn from much fewer examples than outside of the overparametrized regime.

I think this is interesting, in particular the observation of the over-parametrized regime. However, I think the work needs major revisions before being considered for publication in Nature Communications, as there are many unanswered questions as well as uncertainties regarding the significance of the work.

Concerns about the content:

- The claims of the paper rely on the ability to interconnect reservoirs and perform a large matrix multiplication for the final layer. However, here this is performed fully offline by combining recorded data. I think that this should be made crystal clear upfront in the paper because the claims in the introduction can be misleading in that regard.

We thank the reviewer for the suggestion. We have rewritten the introduction and abstract to make it clear what has been performed in experiment and what has been performed offline.

To be clear, we have pseudo-connected the arrays in experiment by measuring the response of the first network layer, selecting a certain frequency channel and converting that to a field input for the second layer. The second layer is then experimentally measured. In the training phase, we combine the responses of each layer offline for training, as the reviewer correctly points out.

- Because this interconnection and matrix multiplication are critical, there needs to be a clear and realistic path towards implementing it in hardware. I fully agree that it makes sense to investigate the computing capability of interconnected reservoirs before launching the difficult hardware implementation. However, I also think that -- in order for this work to be significant -- there needs to be a more rigorous evaluation of how the hardware implementation would work. I detail my questions about it below.

We thank the reviewer for raising this point and for the opportunity to discuss scalability. We initially did not include this as the focus of the manuscript was to demonstrate how to connect / combine physical reservoirs and explore the enriched computational properties by doing so, in particular the result of realising an overparameterised regime. This was not clear before. We have now included a discussion of the scalability of our computing scheme, and discussed ways that the networking method can be applied to any physical system.

- The input for now is a magnetic field. How would this work in a fully hardware implementation? How does this limit the parallelism, compactness, speed and energy efficiency? Do the authors have in mind alternatives to a field input? If yes, what are these and how would they perform regarding the metrics mentioned above.

The required combination of metrics such as parallelism, compactness, speed and energy efficiency varies between different machine learning tasks and applications. For example, a wearable health device that monitors insulin levels and decides on actions based on those levels would need to be compact and energy efficient but would not necessarily require large parallelism or high speeds. Conversely, a neuromorphic chip operating in a data centre that prioritises high performance may require high speed and parallelism to process GHz data from connected components but may not need to be compact or energy efficient (provided that the energy is less than a software neural network performing the same task). As such, benchmarking is often difficult as it's highly dependent on the task and application one is considering. Below we discuss various input methods and provide examples of applications where this method may be useful.

We will focus on input schemes that can switch nanomagnetic states, providing computation that operates in the same way as the scheme in manuscript. We have used global magnetic field input to test the computational capabilities of ASI without the need for complex multi-stage fabrication. However, this method is unsuitable for device integration as the electromagnet occupies a large spatial footprint, the magnetic field is not spatially confined and is prone to disturbing surrounding information, it cannot apply simultaneous inputs to multiple arrays, it has a large energy cost associated with the size of the electromagnet pole pieces and it is limited to kHz speeds.

Magnetic field input can be implemented in a number of scalable ways. Heat-assisted magnetic recording (HAMR) combines a plasmonic transducer and an accompanying electromagnet to store information and is expected to become a key data storage technology in large data centres in the coming years. It may be desirable to combine computing and data storage into the same system. As such, the electromagnet write head designed for use in HAMR could also be used to drive collective state switching locally, allowing for data to be input to multiple arrays on a dense chip. However, electromagnets still have several limitations. In terms of parallelism, 1D magnetic fields are only able to apply a single input at each time step (encoded in the strength in the magnetic field). This can be increased to 2 or 3 simultaneous inputs if the global field can be applied in 2D or 3D (information encoded as magnetic field strength and angle of applied field). Furthermore, the power consumption of electromagnets is large, typically in the 100s mW range making them unattractive for applications with stringent power / energy requirements.

One avenue to continue using magnetic fields but reduce their spatial footprint and power consumption is to fabricate striplines whereby electrical current can induce an oersted field, in turn, switching the nanomagnets. Such striplines can operate using ~ns pulses improving data input speed. At present, we are using mm-scale arrays to improve readout signal to noise, however, suspect that a similar level of performance can be achieved with a ~5 X 5 um array (~100 elements). If a nanomagnetic array were fabricated on top of a microstrip at a separation of 10 nm, the current required to generate a 25 mT field can be is calculated to be 200 mA (details in SI note 11 and the manuscript). This gives a power consumption of 13.4 mW if current is continually applied. This is on the same order of magnitude as other nanomagnetic neuromorphic schemes based on MTJ's [Ross, Andrew, et al. "Multilayer spintronic neural networks with radiofrequency connections." *Nature Nanotechnology* (2023): 1-8.]. The required field, and therefore power consumption, can be reduced by reducing the coercive fields of the nanomagnets either by

modifying their shape or by changing their material. Furthermore, multiple striplines can be fabricated across an array to increase the number of parallel inputs that can be applied.

We can remove magnetic fields entirely by moving to current / voltage induced switching mechanisms. Spin orbit torque switching can be realised by patterning nanomagnets on top of a material with high spin-Hall angle such as Pt, Ta or W. There are numerous examples demonstrating spin orbit torque switching of in-plane nanomagnets such as [Liu, Luqiao, et al. "Spin-torque switching with the giant spin Hall effect of tantalum." *Science* 336.6081 (2012): 555-558., Fukami, Shunsuke, et al. "A spin-orbit torque switching scheme with collinear magnetic easy axis and current configuration." *Nature Nanotechnology* 11.7 (2016): 621-625.] where switching was achieved with current densities on the order of 10^{11} A/m², albeit at a much thinner nanomagnet dimension than the 20 nm thickness used in this work. With some engineering of nanomagnet sizes and materials, we are confident that we can achieve the switching functionality required for computation whilst reducing spatial footprint & energy cost and increasing the input speed of our scheme. We have included a calculation of the power consumption of device utilising spin-torque switching in the manuscript and found the power consumption to be 0.17 mW.

We are currently working on methods of increasing the dimensionality of our input to improve parallelism which will be the subject of a future publication. An example of how this may be achieved is to pattern multiple striplines which can be addressed independently, to selectively switch certain regions of the array.

- The overparametrized regime is here reached with a few thousand parameters. This requires a matrix multiplication of that dimension. Here it is implemented in software. How do the authors plan this in a real system? For real life tasks, I suppose that this size will even go up. As large matrix multiplication is usually the bottleneck in terms of time and energy consumption for neural networks, doing it in software seems limiting to me.

In Figure 1 of the old manuscript, the reviewer is correct that the overparameterised regime is reached at ~1000 parameters. Previously, we sequentially added outputs in a specific order so that we can evaluate how the individual networks behave. During the first 300 parameters, only outputs from the first layer of the PNN are selected (i.e. we are only using outputs from a single physical system). During the next stage (300-1000), outputs from the 2nd layers in the series architectures are randomly added, followed by the third layers. As such, many of the ~1000 parameters required to reach overparameterisation in this case are redundant outputs from the single layer arrays.

We have now explored overparameterisation in more detail and found that minimum number of parameters needed for overparameterisation is far smaller. To reach overparameterisation there are three key parameters: the number of model parameters which here corresponds to the number of weights in the readout layer and therefore the number of FMR channels, the size of the training dataset and the effective dimensionality of the readout data i.e. how correlated are the FMR [Schaeffer, Rylan, et al. "Double Descent Demystified: Identifying, Interpreting & Ablating the Sources of a Deep Learning Puzzle." *arXiv preprint arXiv:2303.14151* (2023).]. In software networks, overparameterisation can be reached as long as the number of model parameters is greater than the size of the training set as network parameters are typically uncorrelated

In our scheme, the effective dimensionality of the readout is less than the number of readout parameters, as FMR channels separated close in frequency are highly correlated (e.g. FMR spectra at 6 GHz \approx 6.02 GHz). Furthermore, many of our FMR channels are dominated by noise where there is no resonance and therefore do not contribute to improved performance.

We have explored the minimum number of channels required for our networks to reach overparameterization by randomly adding channels and evaluating the performance when predicting 6 Mackey-Glass future steps ($t+1,3,5,7,9,11$) of the Mackey-Glass equation. We repeated this 50 times and plotted the mean (solid line) and 2σ (shaded region) train and test MSE as a function of the number of FMR channels.

Figure 1: Neuromorphic overparameterisation. Train and Test MSE for selected architectures when predicting future values of the Mackey-Glass equation when varying the number of trainable parameters (i.e. FMR output channels) for training set lengths of a) 50 and c) 100. MSE is the mean of 50 random combinations of outputs across 6 different prediction tasks. b,d) Improvement factors when moving from underparameterised regime to overparameterised regime. Each network shows three regimes: an underparameterised regime (UP) when the number of parameters is small, an overfitting regime where the number of parameters is near the size of the training set and an overparameterised regime (OP) when the number of parameters exceeds the size of the training set. Example train and test predictions for MS->WM->PW in the underparameterised regime (c), overfitting regime (d) and overparameterised regime (e).

Starting with single systems, we see a rapid decrease in train MSE and a double descent curve in the test MSE which peaks when the number of output parameters equals the size of the training set. However, the MSE in the overparameterized regime is higher than the underparameterised, as such, the system is overfitting i.e. reaching the overparameterised regime here is not 'useful'. For the networked architectures, we see the same initial behaviour, however, the MSE eventually drops below that of the underparameterised regime. The system has entered a beneficial overparameterized regime. As such, the minimum number of parameters required to reach a beneficial overparameterised regime is 100-200 outputs. We would like to point out that this method does not produce the lowest possible MSE for a given task, hence why the MSE's displayed are higher than those presented in other parts of the manuscript. Further details of the overparameterised regime are provided in the manuscript and later in this response.

To achieve matrix multiplication in hardware, we can use memristor arrays to perform vector-vector multiplication. As shown above, we would require ~ 100 s of memristors to perform the multiplication required in the overparameterised regime. As the reviewer states below, these are

routinely made and only consume ~0.1 - 1 pJ of energy per operation at today's standards [Amirsoleimani, Amirali, et al. "In-Memory Vector-Matrix Multiplication in Monolithic Complementary Metal–Oxide–Semiconductor-Memristor Integrated Circuits: Design Choices, Challenges, and Perspectives." *Advanced Intelligent Systems* 2.11 (2020): 2000115.]

- The introduction puts forward the claim that the proposed system offers a "high-dimensional data output" while "Neuromorphic systems tend to employ one or < 10 physical output channels". Memristive crossbar arrays are made with over 500 outputs routinely and state of the art phase change memory chips have tens of millions of trainable parameters. The most recent and most impressive example is Ambrogio, S., Narayanan, P., Okazaki, A. et al. An analog-AI chip for energy-efficient speech recognition and transcription. *Nature* 620, 768–775 (2023) but there are many others. So, I do not think that this claim is valid.

We thank the reviewer for this point. This has now been removed from the manuscript.

Furthermore, in the proposed system, the multiple outputs are for now fully realized by offline post treatment. How would multiple outputs work in a hardware implementation? It seems that it would require either sending a single frequency signal at a time (sequential) or one multi-frequency signal (parallel). Which option is considered and how would it be implemented (signal generation, frequency selection etc.)? If the parallel option is considered, are there any studies about the ability of artificial spin ice arrays to react to multiple signals simultaneously?

In our manuscript, we measure each output sequentially via an rf-diode that converts the AC signal to a DC voltage. We can pattern our arrays in contact with a heavy metal to perform spin-torque FMR, where a magnetisation precession generates a DC voltage in the heavy metal underlayer [Jungfleisch, Matthias B., et al. "High-frequency dynamics modulated by collective magnetization reversal in artificial spin ice." *Physical Review Applied* 8.6 (2017): 064026.] - removing the rf-diode component. As this method is sequential, each output must be temporarily stored prior to processing. One could store these outputs with an accompanying memory cache, or through capacitors / memristors if combined with a relay circuit. This information can then be propagated through a memristor array for inference as previously described.

Parallel readout of all frequencies simultaneously can also be implemented. RF noise sources are available at frequencies near those required in this work (material optimisation can be used to change the operational frequency of our arrays). ASI is routinely simulated in the parallel regime, where all frequencies are simultaneously excited in one frequency step by applying nanosecond field pulses. Provided that the signal is in the linear regime, the rf response of the single and parallel schemes will be the same. The resulting signal could then be, for instance, passed to multiple magnetic tunnel junctions (MTJ) acting as rf diodes with varying response frequencies to simultaneously read the entire frequency response. This signal can then be passed to a memristor array [Ross, Andrew, et al. "Multilayer spintronic neural networks with radiofrequency connections." *Nature Nanotechnology* (2023): 1-8.].

Another option is to micropattern multiple microwave resonators which are tuned to different frequencies, thus simultaneously reading out the frequency response of the array with the added benefit of having spatial as well as frequency dependent information.

Finally, there has been recent work using the magnetoresistance of different artificial spin ice states [Hu, Wenjie, et al. "Distinguishing artificial spin ice states using magnetoresistance

effect for neuromorphic computing." *Nature Communications* 14.1 (2023): 2562.]. One can pattern multiple contacts and measure the magnetoresistance at different spatial locations.

- Exactly what are the trainable parameters and how they are counted is not very clear. How independent are they? How would this change when in a real hardware system with constraints on the input/outputs. In a fully hardware implementation, how would these parameters be stored? Do the authors envision offline or online training?

The trainable parameters here are the number of FMR channels used in the regression step. In our system, the channels closely separated in frequency are highly correlated whereas those further apart are less so. We do not account for this correlation when defining the number of trainable parameters. In a fully hardware system (i.e. with the memristor arrays mentioned above), the number of trainable parameters would again refer to how many output channels you include in the regression step. As mentioned previously, one could store these outputs in a temporary memory cache, or through a set of memristors.

The type of training is highly dependent on the task and the robustness of the system to the external environment. For certain tasks, an initial offline regression step may be sufficient, after which the system constantly runs in inference mode. Linear and ridge regression have closed-form solutions which require matrix-matrix multiplication and inverse matrix operations. Implementing these algorithms directly in memristor hardware has been achieved [Sun, Zhong, et al. "One-step regression and classification with cross-point resistive memory arrays." *Science advances* 6.5 (2020): eaay2378.]. Alternatively, training could be achieved by setting weights with an accompanying CMOS control unit, such as a raspberry pi or Arduino capable of simple programming, after which the system can then run in a low-power inference mode.

It is likely that changing external conditions (e.g. change in input conditions) will require some online updates to the weight matrix. We have demonstrated that our system is able to rapidly adapt to new tasks in an online fashion via meta-learning. Implementing this in would likely require external an CMOS control unit to update the weights. If one could implement some feedback between the predicted output and the stored weights, where the weights are modified to minimise a loss function, then it may be possible to train a system without software.

- At equal number of trainable parameters, how do the different systems (single reservoir, multiple reservoirs) compare? For instance, Figure 3c compares different reservoir types but the number of parameters and number of reservoirs are not addressed clearly. In Figure 3b, the "PNN" has better accuracy than the other systems, but also has the largest number of nodes. How does the nodes number relate to the parameters number? At equal number of nodes, do the other systems perform as well as the PNN?

The reviewer is correct that the PNN has a far greater number of parameters than the other networks. One of the key advantages to the networking approach, in particular the PNN, is the production of a large set of diverse parameters. This is key for achieving strong performance and reaching a beneficial overparameterised regime. We could arbitrarily increase the number of parameters for a single array by increasing the resolution of the microwave readout, however, this would not improve performance as the additional outputs would be highly correlated.

We have now commented on the number of outputs used in the main text as well as SI note 8. For the results presented in the main text, the number of outputs used for each architecture are as follows: WM - 34, MS+PW+WM - 44, MS->WM - 42, MS->WM->PW - 93 and PNN - 13567. The number of nodes for the PNN is far greater than the other architectures. This is because the

feature selection algorithm naturally evolves to putting the PNN in the overparameterised regime as this is where performance is best.

If we restrict the PNN to operate in the underparameterised regime we see that MSE increases.

Figure 2: MSE profiles for a) Mackey-Glass future prediction, b) NARMA transformation and c) future prediction of NARMA-7 processed Mackey-Glass for the best single, parallel, series and PNN when the PNN is forced to be underparameterised (i.e. the number of parameters is less than the size of the training set). PNN MSE is similar to the series networks as it is not able to harness all of the outputs.

For some tasks, the MSE is worse than the smaller networks. This is because the feature selection algorithm is stochastic and requires more iterations to explore the entire readout space of the PNN in comparison to the simpler networks. Additionally, the algorithm finds a set of outputs that performs well at all tasks. Given enough iterations, the performance of the PNN will at least match other architectures, as the PNN contains outputs from all other network architectures.

- More generally, the proposed system is not rigorously compared to conventional hardware in terms of energy efficiency / size / speed. Thus, it is hard to assess the relevance of the work.

We have now included a section on scalability incorporating the above points, assessing the power consumption of a device and we believe the manuscript has greatly improved as a result.

- What sets the number of parameters to reach overparameterization? Does it depend on the task, number of examples, network? For different architectures of reservoir systems, is it reached at the same number?

We have now performed a more detailed analysis of overparameterisation. We saw in the previous response that the size of the training set does play a role, with more data requiring more parameters to reach overparameterisation. To illustrate how architecture influences overparameterisation, we have analysed the MSE vs P for three series networks: MS-→ MS, WM-→ WM and PW-→PW.

Figure 3: MSE dependence on the number of trainable parameters for network with one physical system. a) MSE curve for 2-series networks comprising one physical system for each reservoir explored in this work. b) MSE curve for physical neural networks comprising one physical system. For 2 series networks, only WM-> WM reaches a beneficial overparameterised regime. For PNNs, all systems can reach a beneficial overparameterised regime.

We see that all three architectures observe the double descent phenomena and reach an overparameterised state. However, for MS->MS, the MSE in the overparameterised regime is higher than the underparameterised regime. In this case, reaching overparameterization hinders performance. For WM->WM and PW->PW, the MSE in the overparameterised is lower than the underparameterised, hence reaching overparameterisation here is useful. The reason for the differences come from the complexity of the reservoir, MS is a fairly linear reservoir with low memory-capacity, hence the dimensionality of the states is limited (i.e. many of the outputs are linear scalings of one another and are redundant during the regression step). Conversely, WM and PW have a mixture of history dependent and non-linear responses, therefore the dimensionality is higher, and the system can reach a useful overparameterisation state.

Figure 4: Effect of task on overparameterisation. MSE dependence on number of parameters when predicting a) $t+5$ and b) $t+11$ of the Mackey-Glass equation as well as transforming a sinusoidal input to c) $\sin^2(x)$, d) $\sin(3x) + \cos(x)$ and e) $\cos(2x)\cos(3x)$. The benefits of overparameterisation are task dependent.

In terms of task, we see similar behaviour across all future time steps for predicting the MG equation. If instead we look at overparameterisation when performing the sine transformation tasks, we see that for tasks that only require non-linearity only, all of the samples reach a useful overparameterised regime. These tasks are much simpler and so this behaviour is not totally surprising. Conversely, for transformations requiring some memory of previous inputs (e.g. sine \rightarrow saw), there are signs of a double descent curve in some arrays, particularly PW, but no individual arrays reach a useful overparameterised regime where MSE reduces. For the 3 parallel and PNN cases, useful overparameterisation is reached for all tasks.

In summary, reaching a useful overparameterisation regime depends on both the task and architecture. Single reservoirs can be overparameterised for simple tasks, whereas more complex tasks that require memory capacity require networks of reservoirs. For networks that display overparameterisation, the peak in MSE typically occurs when $N = P$, the final MSE, and whether this is lower than the underparameterised regime depends on the architecture. The point at which the overparameterised MSE goes below the underparameterised is related to task complexity and network architecture. A more thorough analysis of this will be the subject of a future publication.

- Is a specific design of the reservoir system needed for each task? Also, how is a design reproducible for different reservoirs that are nominally identical, i.e. is device variability a problem? How would this design work in applications? Are there possibilities to generalize? How does this work relate to other works on interconnecting reservoirs (such as L. Grigoryeva, J. Henriques, L. Larger and J. Ortega, "Nonlinear Memory Capacity of Parallel Time-Delay Reservoir Computers in the Processing of Multidimensional Signals," in *Neural Computation*, vol. 28, no. 7, pp. 1411-1451, July 2016, doi: 10.1162/NECO_a_00845.)?

In general, single physical reservoirs often have a fixed set of internal dynamics and therefore only perform well at certain tasks. Until this work, if one desired to perform well at a different task, or perform well across multiple tasks, then multiple reservoirs would have to be fabricated. The advantage of the networked approach is that reservoirs can be combined to perform better than the sum of their parts e.g. no individual reservoir performs well at MG $t+5$, but when they are networked together, the performance substantially improves.

We have not explicitly tested device variability, however, we believe that it will not be a concern. Here we rely on the collective response of many individual elements. Each element will have slightly different dimensions and responses to external input but when averaged over many elements, the variability reduces. In fact, variability in the individual elements benefits here. If all elements were to switch at the same input step, then we would not be able to implement our computing scheme. In practice, any variability in devices would lead to a slightly different input range being required and slightly different resonant frequencies. A simple calibration step after fabricating the device to determine the appropriate input range and measurement frequencies would be sufficient to correct for this variability.

When selecting which output features to use for a given network, we have designed the feature selection algorithm to find the best set of features across all tasks, rather than in a task-specific way i.e. the final set of features performs well when predicting all future steps. Therefore, once this feature set is found, only the training of weights for a specific task is required rather than having to search through all features again. In that sense, the networks presented in this work are already general.

In terms of network architecture, here we have optimised output / input connections from one reservoir to the next to maximise memory capacity, which gives good general performance when predicting future steps of the Mackey-Glass equation. We could instead optimise for non-linearity, in which case a separate set of output / input connections would be required. One could also train the output / input connections to give task-specific performance or good general performance across multiple tasks. We are currently exploring this avenue of research which will be the subject of a future publication.

The PNN in the manuscript utilised 45 different series networks which would require the fabrication of 99 distinct arrays if all the responses were to be simultaneously measured. Reaching overparameterisation does not require this many arrays. In principle, one could construct a 3 x 3 network of arrays and achieve similar performance to that described in this work. If time and memory were not constraints, then this could be reduced to just one copy of each reservoir provided the reservoir states can be temporarily stored, as done in this work.

In relation to other work, as mentioned in the manuscript, parallel networks have been implemented in both software and hardware. This forms the basis of the 'virtual node' method used to artificially enhance the number of reservoir outputs by feeding a masked input multiple times to the same reservoir. Series networks have been implemented in software but translating this to hardware is non-trivial due to the vast number of possible interconnections. We have provided a methodology of interconnecting physical systems as well as exploring the overparameterised regime that arises from doing so.

Concerns about the presentation:

- The paper is quite dense and honestly it took a lot of work to understand what was actually done. It would really help the readers, especially those not in spintronics or spin ice community, to get a clear description of the system and results and move more technical details to methods and Supplementary.

- Figure 3 is confusing. Too many acronyms are used, that are not legible to outsiders of the subfield. The color code is unclear.

In conclusion, the scientific findings of the study seem sound to me but:

- There are things left to clarify in the over-parametrization study
- Without a better understanding of how the concept would work in a real system and without comparison to conventional hardware, it is difficult to grasp how significant this paper is.
- The claims about output dimension superiority do not seem justified to me.
- The paper's clarity must be improved

We thank the reviewer for these suggestions. The manuscript has now been overhauled and we believe it is now much clearer.

Reviewer #2 (Remarks to the Author):

The authors have studied the multilayer physical neural networks (PNNs) based on three types of artificial spin ices (ASIs). The authors quantified the memory capacities and nonlinearities for these three ASIs and benchmarked their performance for Mackey-Glass and sine transformation problems. Then, the authors constructed various types of PNNs based on different combinations of these three ASIs and benchmarked their performance. In the end, they show that the multilayer PNN can realize a double-descent curve and enable few-shot learning. Overall, this study is comprehensive and interesting as it is the first study using non-CMOS technology to show the possibility of realizing an over-parameterized regime. However, I have serious concerns about the benchmark and reproducibility of this work as the key novelty and significance are in the system (algorithm) design, which is highly dependent on the details. If the authors can address these two issues, I am happy to support its publication.

Benchmark issues:

1. Although this paper claims that the PNN with ASIs can realize an over-parameterized regime, it still requires significant resources, including both ASIs and supporting CMOS circuits to process signals between layers. In contrast, CMOS-only multilayer neural networks (software-based) can also realize this over-parameterized regime as refs. 73, 75, and this paper: <https://arxiv.org/abs/2303.14151> show. This suggests the importance of carefully benchmarking the resources to achieve the over-parameterized regime.

As mentioned in the response to reviewer 1s queries about overparameterisation, Figure 1 of the original manuscript does not represent the minimum number of features to reach overparameterisation. In our response to this query, we analysed the minimum number of features needed to reach overparameterization and found that it can occur when the number of parameters is equal to the number of training data points, provided that the network is sufficiently complex for the task (e.g. a single reservoir for nonlinear tasks, or a 2 series network for tasks requiring memory).

We would like to point out that the prediction tasks performed in this work require memory of previous inputs. For this, a conventional MLP would perform badly as there would be no way to recall past information (see SI note 9 for more details). Furthermore, the tasks often performed in

literature are classification tasks, whereas we are performing regression. This makes a concrete comparison between existing literature and our scheme difficult.

2. While ASIs can realize the over-parameterized regime, nonlinearity is not required at the fundamental level. It has been shown that a double descent curve can happen in ordinary linear regression (<https://arxiv.org/abs/2303.14151>).

We thank the reviewer for sharing this paper. In that work they show double descent curves occurring with linear regression, but the degree to which it is useful is not entirely clear. For example, in Figure 2 of that work (pasted below), in the overparameterised regime, the model fits the training data perfectly but does not generalise well to unseen data. Furthermore, the MSE does not reduce significantly in the overparameterised regime compared to the underparameterised regime. This is also evident in later figures of that paper e.g. Figure 3 where the MSE in the overparameterised regime (<17 training data points) is far greater than the underparameterised regime. As such, we must make a distinction between useful and non-useful overparameterisation as we have done in our response to reviewer 1.

Additionally, in that work they observe double descent when using linear regression. However, their dataset is multidimensional. Here we are inputting a 1D time series. Any linear transformation of this input series can only scale the input data, irrespective of the number of parameters. Therefore, in the scheme we present, some degree of nonlinearity is required.

Figure 2: Intuition for double descent from polynomial regression. Top: Polynomial regression displays double descent. Bottom: When *underparameterized*, the model is unable to capture finer-grained features in the training data, meaning bias is large but variance is small. As the interpolation threshold is approached, the training data can be fit exactly, meaning bias is small; however, the particular realization of the training data significantly affects the learnt function, meaning variance is large. When *overparameterized*, the model can exactly fit the training data, meaning bias is again small, but the model is also regularized towards a small-norm solution, making variance small.

- How does a PNN with ASIs perform compared with conventional CMOS systems (software-based) for these simple tasks?

We have now performed a more rigorous comparison to software networks which we have included in the supplementary information.

Figure 5: Comparison to echo state networks. ESN performance when varying the number of ESN nodes for a) Mackey-Glass future prediction, b) NARMA transform, c) NARMA transform plus future prediction as presented in the main text. In each plot, the blue line and shaded region represent the average MSE and error over 50 randomly initialise ESNs at a given number of nodes (N_{nodes}). Solid flat lines represent the MSE from the physical networks presented in the main text. Here, the training data size (N_{train}) = 200. MSE is an average over multiple tasks: $t+0$ to $t+12$ for future prediction (a), NARMA 0 - 12 for NARMA transforms (b) and $t+0$ to $t+20$ for NARMA plus future prediction (c). Panels d-f) show heatmaps of MSE for varying N_{train} and N_{nodes} .

Figure S12 (above) shows the performance of ESNs with varying number of nodes (N_{nodes}) for a) Mackey-Glass future prediction, b) NARMA transform, c) NARMA transform plus future prediction as presented in the main text. Panels d-f) show the ESN performance when varying N_{train} and N_{nodes} for each task. In a-c), the blue line represents the average error of 50 randomly initialised ESNs for a given number of internal nodes with shaded area representing $\pm 2\sigma$. Coloured lines show the performance of the various networks explored in the main text. In a-c), N_{train} = 200. MSE represents the average performance over multiple tasks: $t+0$ to $t+12$ for future prediction (a), NARMA 0 - 12 for NARMA transforms (b) and $t+0$ to $t+20$ for NARMA plus future prediction (c). For ESNs, as N_{nodes} and N_{train} increases, MSE reduces as expected. For each task, we find that single arrays are matched by ESNs with 20 - 40 nodes. As the physical network complexity

increases, the corresponding ESN size to match performance also increases. The ESN size required to match the PNN performance is 100 for the NARMA task and >500 for prediction tasks.

Figure 6: Comparison to multilayer perceptrons (MLP). MLP performance when varying the number of hidden nodes (N_{hidden}) and the number of previous inputs given to the model during training (T_{seg}) for a) Mackey-Glass future prediction, b) NARMA transform, c) NARMA transform plus future prediction as presented in the main text. In each plot the blue line represents the MSE of the MLP for a given number of hidden nodes. Solid flat lines represent the MSE from the physical networks presented in the main text. Here, the training data size (N_{train}) = 200. MSE is an average over multiple tasks: $t+0$ to $t+12$ for future prediction (a), NARMA 0 - 12 for NARMA transforms (b) and $t+0$ to $t+20$ for NARMA plus future prediction (c).

We now compare the physical network performance to multilayer perceptrons (MLP). MLPs are static, i.e. they hold no information about previous inputs and will fail at predictive tasks. As such, we vary the number of previous inputs given to the MLP during each time step. Figure S13 (above) shows the performance of MLPs when varying the number of hidden nodes (N_{hidden}) and the number of previous inputs provided to the MLP (T_{seg}) for a) Mackey-Glass future prediction, b)

NARMA transform, c) NARMA transform plus future prediction as presented in the main text. As with the ESN, MSE represents the average performance over multiple tasks: t+0 to t+12 for future prediction (a), NARMA 0 - 12 for NARMA transforms (b) and t+0 to t+20 for NARMA plus future prediction (c). The MLP MSE reduces as N_{hidden} and T_{seg} increases as expected. For $T_{\text{seg}} = 1$, the physical networks outperform all trialled MLPs due to the lack of memory of previous inputs in the MLP. We find that series networks are well matched to MLPs with $T_{\text{seg}} = 2$ and $N_{\text{hidden}} \sim 10$ and the PNN is matched to an MLP with $T_{\text{seg}} = 3$ and $N_{\text{hidden}} \sim 10-50$ for all tasks. Whilst the resulting MLP is simple, the network interconnections must be trained using gradient descent. This is inexpensive for small networks, but as network and task complexity increases, so does the expense of training MLP weights.

- What is the key point for the emergence of this double descent curve? Can you use PNN with only one ASI to achieve this double descent curve? If not possible, why? (In my opinion, it should be possible as simple CMOS-only software can achieve it.)

As mentioned in our response to reviewer 1, there are several factors that influence whether double descent and a useful overparameterisation regime is achieved: the number of training data points, the number of model parameters, the effective dimensionality of the readout layer and the task you perform. For sine transformations, a single ASI is sufficient to reach a useful overparameterisation regime where MSE reduces below the underparameterised regime. For memory dependent tasks, the single ASI's on their own are not sufficient. Whilst we observe double descent phenomena, the MSE in the overparameterised regime is higher than the underparameterised regime. That's not to say that all single reservoir systems are insufficient, only that the reservoirs that we have tested do not possess the necessary dynamics and dimensionality to reach a beneficial regime. If one were to build a physical reservoir with sufficient memory capacity and non-linearity, then this could reach an overparameterised state provided the readout had sufficient dimensionality.

Below we show two cases of incorporating the same ASI into a larger network.

Panel a) shows 2 layer series networks comprising a single array. We see that networks with WM and PW samples sufficient to reach an overparameterised state when predicting t+5 of the

mackey-glass equation. Whilst we have not tested it, we suspect that given enough layers, a series network comprising only MS will reach a useful overparameterised state. In panel b) we have tested numerous networks where MS forms the second layer but takes an input from another sample. If we extract all of the readouts of the MS sample and exclude the first layer from which the input came, we can see how a PNN comprising only MS would perform. We observe that the system reaches a useful overparameterisation regime.

As such, we conclude that a single ASI can reach an overparameterised state provided that the network is sufficiently large to produce the complex readout dynamics required. More generally, any single reservoir can be used to reach overparameterization provided it has some degree of non-linearity and, for prediction tasks such as predicting the Mackey-Glass equation, some degree of memory capacity.

- If one can use PNN with only one ASI to achieve this double descent curve, what is the performance in the over-parameterized regime?

Please refer to the above response

Reproducibility issues:

1. The authors should show tables to indicate the details of every neural network being used in Fig. 1, Fig. 3, and Fig. 4
2. I am highly concerned about the reproducibility of this work. The results are highly dependent on the details. Data and codes should be made publicly available to enable reproducibility:
 - All figure data for plotted figures
 - Pseudo-code for hardware/software co-design system should be provided in supplementary and the comments of the submitted codes.
 - All codes with raw data for hybrid hardware/software experiments. I understand that intermediate experimental results may not be available. Then, the software codes to collect data and post-process should be shown.
 - All codes for purely software simulation to ensure the reproduction of figures like Fig. 1, Fig. 3, and Fig. 4

We thank the reviewer for these suggestions. The manuscript has been overhauled and we believe it is much clearer. A table has been added to the supplementary information detailing which networks have been used for each figure. These refer to the available data which can be found at “Github.com/StenningK/NeuroOverParam”. We have included all codes and data used to reproduce the findings of this manuscript as well as pseudocode for all non-standard algorithms in the Methods section. To be clear, none of our results are purely software simulation. All the data is recorded experimentally, only the training of weights takes place offline.

Reviewer #3 (Remarks to the Author):

The paper proposes using connected artificial spin ice arrays of different geometries for neuromorphic computing. It studies how macrospin square, width-modified square and gradient pinwheel geometries affect the response and explores different ways of connecting these reservoirs into a multi-node network. I find the paper relevant and timely, and the results original and useful to the community. At the same time, I think that it should be more accurate or detailed in several key aspects. If these are properly addressed, the paper can be recommended for publication and will be of interest to the wide readership of Nature Communications.

We thank the reviewer for their positive appraisal of our work and their recommendations for improvement. The manuscript has significantly improved as a result.

1. “Neuromorphic schemes typically employ a single physical system, limiting the dimensionality and range of available dynamics – restricting strong performance to a few specific tasks” is a sentence from the abstract that is false. The field of neuromorphic computing is broad and includes some very sophisticated systems (e.g. <https://doi.org/10.1038/s41928-022-00838-3> analog reservoir with output layer; True North architecture from IBM; <https://doi.org/10.1038/s41427-021-00282-3> all-spin networks, etc.). Same applies to the other instances of using “neuromorphic” in the paper, for instance in “tend to employ one or ≤ 10 physical output channels”. Perhaps, replacement with “physical reservoir computing”, which is a small subset of neuromorphic computing, would make it more precise.

We thank the reviewer for this suggestion. The reviewer is correct that “Neuromorphic schemes” is too broad a category for the claims we have made in the abstract and introduction. This has been replaced with ‘physical neuromorphic computing’ throughout.

2. I struggled to find the boundary between experiments and simulations in this work. If every task has been applied to every architecture experimentally, it should be clarified. If the response of each type of reservoir has been measured once and then fed into a simulation, it should also be explicitly stated.

We thank the reviewer for raising this point. To be clear, we have experimentally measured the response of every network architecture to a given input and then concatenated this response offline. No responses have been simulated. When performing future prediction tasks, a new output matrix is trained for each future step i.e. $t+1$, $t+2$ etc. We have added this into the manuscript.

3. The PNN seems to be the pinnacle of the work, yet its exact structure remains unclear apart from its composition of 52 sub-networks. I could find the information on feature selection in the PNN but not on how and why it has been designed in a certain way.

The PNN comprises the responses of every subnetwork architecture that we have explored. When testing series networks, we created 49 different networks. For the PNN, we have combined the response of each of these networks offline. This has been made clear in the text.

4. Better understanding of the PNN’s complexity and composition is also relevant for comparing its performance to simpler counterparts in Figure S6b. It seems that a significant increase in complexity leads to a rather marginal performance improvement or none (in 50+% cases). Thus, having PW, PW+WM, WM->PW (5 ASI in total) and selecting one depending on the task outperforms PNN (106 ASI) on average. This raises a question: what is the relationship between performance improvement and added network complexity? Would a 15-node PNN yield similar results, or would a 150-node network significantly enhance performance?

Figure S6. a) MSE for single and parallel configurations when transforming a sine input. Improvements are obtained across asymmetric (memory-capacity + nonlinearity) tasks, up to $4.4 \times$ for $\sin(3x) + \cos(x)$. Performance often worsens for symmetric (nonlinearity only) tasks. b) MSE profiles the best single, parallel, series and PNN networks when transforming a sine input. Series improvements are only observed when connecting WM and PW for more complex asymmetric tasks. PNN outperforms other architectures for 9/20 tasks.

The reviewer is correct that for the sine transformations, the PNN only improves on 50% of the cases. We would like to make it clear that here, the feature selection algorithm selects features that perform well across all tasks, rather than a specific task, as such the average performance across all tasks is lower for the PNN. If we were to train for specific task, we would see the PNN have the best performance for all tasks, as the PNN contains each of the subnetworks. The same method is used for the Mackey-Glass prediction and NARMA transformations throughout the main work.

5. Continuing question 4, what is the suggested optimal complexity for an individual node? To what extent should these multi-reservoir networks shift toward RNNs to maximize benefits while still maintaining relatively straightforward training? This is particularly important for hardware systems, in which interconnects between reservoirs are “expensive” and complex due to a combination of CMOS and non-CMOS components, such as ASI.

For strong performance across all tasks, i.e. both non-linear transformations and predictions, an ideal node would perform linear and non-linear transforms of current and past input data. If we could hypothetically split the node dynamics into different portions, one portion would be dedicated to linear transformations of the current input data, one dedicated to non-linear transformations of current input data, then portions dedicated to transformations of $t-1$, $t-2$... as well as mixing of previous inputs i.e. a transformation of $t-1$ and $t-2$ at the same time. Ideally, the range of non-linear transforms would be large such that the Fourier components of these transforms cover all frequencies. Furthermore, these components would be all separated in readout space such that the regression can access all the dynamics, and only use readout channels suitable for the task at hand (e.g. for a symmetric transformation, the regression would ignore all channels which transform previous inputs). Of course, creating such a node and reading out these distinct behaviours is not trivial. Physical systems often have a limited range of internal dynamics. As such, we propose to build PNNs, where different physical systems perform subsets of the previously mentioned functionalities and are interconnected according to the task at hand.

Regarding RNNs, reservoirs themselves are RNNs with the additional benefit of not having to train the internal connections – a challenging and time-consuming task in software networks. Shifting more towards trainable RNNs could be attractive provided that the training is simple. For

example, the WM sample consists of wide bars which switch and thin bars which remain fixed. We showed in our previous paper that the magnetisation direction of the thin bars affects the system dynamics [Gartside, Jack C., et al. "Reconfigurable training and reservoir computing in an artificial spin-vortex ice via spin-wave fingerprinting." *Nature Nanotechnology* 17.5 (2022): 460-469. – See Figure 2b]. As such, it may be possible to train the dynamics of the array by switching certain fixed bars. This could be achieved through all-optical magnetic switching [Stenning, Kilian D., et al. "Low-power continuous-wave all-optical magnetic switching in ferromagnetic nanoarrays." *Cell Reports Physical Science* 4.3 (2023)], however, knowing which bars to switch would require sophisticated learning algorithms that do not rely on backpropagation.

6. The comparison to a software ESN in Figure 3b is ambiguous. The activation function f is undefined. Assuming that it is something typical like \tanh , such a 90-node single-reservoir ESN is fundamentally simpler than a 100+ ASI multi-layer PNN with a) much more complex nodes, b) trainable weights between the nodes. This result can be used as a reference but not as a direct comparison and not to claim "significantly better performance".

We have now performed a more comprehensive comparison to software. Please see our response to reviewer 2 for details.

7. Dimensionality of a physical reservoir and dimensionality of its output are used interchangeably in the discussion of dimensionality in the bottom section of page 1. If a system has 6 physically fabricated readout electrodes, it does not mean that its dimensionality as a reservoir is 6 (it can be anything) but rather that the output dimensionality is 6 (assuming the readouts are not linearly dependent). The FMR-ASI may have a high dimensionality, however, the readout from these dimensions is still done in series (frequency-by-frequency). Therefore, I suggest correcting the statement and answering the question of how the FMR readout dimensionality of the used ASI can be improved.

We thank the reviewer for this point. This has now been updated. We have included a discussion of how to implement parallel detection in the 'Scalability' section of the manuscript.

8. There must be some confusion about MC and NL metrics. For example, MC of PW is 3 on Fig. 2i (lower panel), 5 on Fig. 2j and "4-7" in the text.

The results in the manuscript are correct. In Fig. 2i (now Fig 2 d), we have analysed the memory and non-linearity of each individual FMR channel, i.e. how correlated is this signal with the previous n inputs. The reason for analysing the response in this way is so that we can see which channels contain as much information about previous inputs as possible. The memory capacity for each channel is the sum of MC_n from $n = 0-7$. We then select these channels as inputs to the next layer. '4-7' refers to the heatmap plotted in Figure 2i (now SI note 4) which looks at which previous channels in more detail. Some channels are more correlated to short term inputs whereas others more correlated to long term inputs. E.g. 9.5 GHz is correlated to steps 1-3 and 8.9 GHz is more correlated to 4-7 time steps.

Finally, in Figure 2j (now Fig 2f) we calculate the memory capacity using all channels. I.e. we can use information from any channel to extract information about previous states.

The text has been updated to make the differences between the different types of analysis clearer.

Related to this is the question of how MC was determined from the black-blue colormaps in Fig. 2g-i. For example, in 2i) at ~ 8.7 GHz the memory is completely lost at steps 2-5 (black) but then recovers at step 6-7 (white). How is it possible? And if it is meaningful, could the system recover its memory at step 100 making its $MC=100$?

Memory capacity is calculated in the same way as other tasks in the manuscript, except that the target is previous inputs. I.e. the target $y\sim(t) = (s(t - \delta t), s(t - 2\delta t), \dots, s(t - k\delta t))$ where s is the driving signal and k ranges from 0 - 8.

We set X (the training data) as the FMR output channels and fit a linear weight matrix between X and y using linear regression. For a single frequency channel, as performed in the heatmaps in (SI note 4), we are scaling the response of that channel and calculating the R^2 between the input and the channel response. In other words, we are looking at the correlation between the channel response and previous values of the input.

The reason for this observed behaviour is that different frequency channels correspond to different subcomponents for our array. The FMR response of an individual element is determined by its shape, magnetisation texture etc. As such we are probing different elements. The PW sample was fabricated with a broad range of nanomagnet sizes. This analysis shows that nanomagnets whose resonance is at ~ 8.7 GHz have a response that is strongly correlated with steps 6-7 but not short term time steps i.e. these elements have a strong fading memory that takes a number of inputs to evolve.

Here we are evaluating MC based on the Mackey-Glass equation which has an underlying periodicity of ~ 22 points. As such, we fix k to be a maximum of 8 to avoid any false memory arising from the correlation of the input with itself. In Figure S3 (pasted below) we vary k_{max} and evaluate the MC. The green curve corresponds to the input correlation with itself. Beyond $k_{max} = 8$, the MC of the input increases due to self-correlation.

Figure 7: a) memory-capacity and b) nonlinearity when varying how many previous inputs are used in the metric calculation. Memory-capacity profiles show characteristic humps and continual rising indicative of correlation between current and past input profiles from the periodic nature of the Mackey-Glass input equation.

9. The authors could add a section discussing the prospects and generalizing their findings including a direction for further improvement, potential problems, what combination of reservoir dynamics could be the most promising, extension to other physical systems etc.

We thank the reviewer for this suggestion. We have added a section on ‘Scalability’, included a discussion on how to achieve overparameterisation in any physical system in the ‘overparameterisation’ section and included open questions and suggestions for improvement in the conclusion.

Nitpicking:

1. “Figure 1, point 1” in chapter Network overview is not clear. Perhaps calling it a region would help.
2. More rigorous language might improve the read, counterexamples include “available “for free”, “extremely high dimensionality”, “our efficient method”, all taken from Conclusions section.
3. Should the equation on page 9 be $y(t) = \text{sum}() + y\text{OFFSET}$?
4. What are the colored traces in Fig. 4c?

These points have been incorporated in the manuscript.

In conclusion, this is an interesting work delving into the topic of multilayer networks comprised of physical reservoirs. It is impossible to imagine real competition between the proposed system and a software network at this point in time. The same, however, can be said about any physical reservoir. This is why I believe that detailed information on the PNN- and how it was constructed, as well as generalizations applicable to alternative physical reservoirs, could be of significant value to the field. This is also why I think that many practical questions (node connectivity, scalability, power consumption, footprint, etc.) are nonessential and can be omitted. A fair comparison to the software counterpart, in this perspective, is useful as a benchmark but not in the sense of direct competition.

REVIEWER COMMENTS

Reviewer #2 (Remarks to the Author):

I appreciate the authors' effort in addressing my concerns. I am fine with the new results based on single ASI systems. However, I am sorry that I cannot agree with the arguments between PNN and CMOS-based systems.

First, the authors didn't address my first comment using solid evidence. Instead, they argue that it's hard to compare. The ASI approach requires CMOS circuits as the interface to process signals between PNN layers. Software-based approaches are also based on CMOS hardware. The only difference is the construction of the reservoir. The authors should clearly benchmark the resources (PPA: power, performance, and area) for ASI and CMOS-based reservoirs. Otherwise, I feel that this is only an interesting concept but will never be useful in practice, making this paper much less attractive as an algorithm/system paper.

Second, the second comment was also not addressed at the hardware level. They should not only use MLP. They should build faithful reservoirs that are similar to PNN using CMOS and then benchmark the resources.

Third, I am not in agreement with a statement like this: "To be clear, none of our results are purely software simulation. All the data is recorded experimentally, only the training of weights takes place offline." In my opinion, the spectra of these ASIs were only taken at a particular time. Then, the authors assumed that these data could be repeatedly used for their "simulation." However, it is not a real experiment due to the following reasons. (1) The spectra can drift significantly over time! So, the data taken at time 0 can not be used for simulation at time T without justification. (2) In all simulation experiments (like running a PNN to achieve over-parametrization), one needs to repeatedly measure the PNN spectra, and thus, actual measurements on the PNN will take a huge amount of time, which is also feasible to me.

Reviewer #3 (Remarks to the Author):

The authors have provided a comprehensive response to the questions raised. My only remaining concern is the broad use of "neuromorphic computing". While it is appropriately replaced by "physical reservoir computing" in most instances, I disagree with its use in the phrase "we present solutions to key outstanding problems in the physical neuromorphic computing field". As mentioned in my initial review, the field of physical neuromorphic computing is much broader than just reservoir computing, and the problems targeted by the authors do not exist in many of its sub-areas.

Once this is addressed, the paper can be recommended for publication.

p.s. there is a typo in: "show allow memory and non-linearity improvements"

Reviewer #2 (Remarks to the Author):

I appreciate the authors' effort in addressing my concerns. I am fine with the new results based on single ASI systems. However, I am sorry that I cannot agree with the arguments between PNN and CMOS-based systems.

We thank the reviewer for their suggestions below. We have implemented the majority of their suggestions, explaining the reasons prohibiting us from implementing all of them. We believe that the manuscript quality has been greatly improved as a result.

First, the authors didn't address my first comment using solid evidence. Instead, they argue that it's hard to compare. The ASI approach requires CMOS circuits as the interface to process signals between PNN layers. Software-based approaches are also based on CMOS hardware. The only difference is the construction of the reservoir. The authors should clearly benchmark the resources (PPA: power, performance, and area) for ASI and CMOS-based reservoirs. Otherwise, I feel that this is only an interesting concept but will never be useful in practice, making this paper much less attractive as an algorithm/system paper.

We thank the reviewer for the suggestion to expand the resource analysis to better situate nanomagnetic systems within the broader computing field. As a preface, we would like to point out that the original motivation for this manuscript was not to suggest that the exact nanomagnetic arrays used here would be the system of choice for scalable neuromorphic computing. Instead, the focus was to demonstrate that physical systems can achieve performance improvements in an analogous way to software networks. While software echo-state networks (ESNs) have been around for some time now, networks of interconnected 'deep' or hierarchical ESNs are a much more recent idea, with the first demonstration in 2017 and much research still ongoing. Until our present work, there was an open question on how well the comparison between software ESNs/reservoir computing and physical/neuromorphic reservoir computing held – is there a relatively loose comparison with a few similarities in operation/performance, or does physical reservoir computing closely embody the behaviour of software ESNs?

One key question that had not been answered is whether creating interconnected 'deep'/hierarchical networks of physical reservoirs leads to the same enhanced performance as for software ESNs, and specifically do the rules governing creating software deep ESNs hold for physical reservoirs, namely that the order of the interconnected ESNs must be arranged in a gradient from short term memory -> long-term memory?

In this study we show for the first time that the comparison between software ESNs and physical reservoirs does indeed hold when interconnecting deep/hierarchical networks of physical reservoirs. We demonstrate enhanced task performance, enhanced computational metrics, and we show that the physical reservoir networks must be ordered in the same short-term to long-term memory arrangement in order to demonstrate enhanced computation.

Alongside this, we show how the same class of physical system, here nanomagnetic arrays, can be tailored to have short term to long term memory via simple tuning of the geometric

parameters, allowing us to physically access the range of memory scales required to construct a useful deep reservoir network.

Beyond this, we show that physical systems can exhibit beneficial overparameterisation behaviour and enhanced learning in an overparameterised regime, which we showcase by implementing few-shot learning. This is the first time that these effects have been shown, and our focus with this paper are demonstrating these conceptual advances to the field and explain how they can be achieved in an arbitrary physical system. It was never our intention to claim supremacy over CMOS or other physical systems.

To directly address the reviewer's comment, we have benchmarked the resources required to operate echo-state networks, which serve as the closest software analogue to physical reservoirs. This benchmarking includes comparisons of power consumption, processing speed, and energy efficiency between our current computing architecture and proposed future devices. These enhancements, along with our expanded commentary on benchmarking and scalability, have significantly improved the quality of the manuscript. We believe these revisions now align with the rigorous standards of *Nature Communications* and hope they fully address the concerns raised.

In what follows, we discuss the updated benchmarking results. These results have been incorporated into the manuscript in 'Discussion' as well as in the supplementary information. In software, an ESN is initialised with random interconnections. Applying input and updating the ESN takes the form of a set of matrix calculations described in the manuscript as:

$$\mathbf{x}(t + \delta t) = (1 - \alpha)\mathbf{x}(t) + \alpha f(\mathbf{W}_{in}\mathbf{s}(t) + \mathbf{W}_{esn}\mathbf{x}(t))$$

Where \mathbf{x} is the ESN state, \mathbf{W}_{in} and \mathbf{W}_{esn} are matrices representing the randomly initialised input and internode weights, α is a scaling factor and f is a non-linear activation function (here tanh). This is repeated for the entire input sequence, with states saved at every input. Following this, regression is performed on the saved states to give a computational output.

For nanomagnetic arrays, a state update corresponds to application of an input (global magnetic field, Oersted field or spin-orbit torque). The array updates itself naturally through the physics. The state is then readout via ferromagnetic spectroscopy and saved via analogue to digital conversion. After this, regression is then performed via conventional CMOS, as it is in the software reservoir.

As such, any differences in performance arise from the input / update process, the readout, and analogue to digital conversion.

We begin by analysing the resources of software reservoirs. A common way of benchmarking the resources of software algorithms is to analyse the number of floating-point operations (FLOPs). We can dissect the reservoir update formula to analyse the number of FLOPs in relation to the number of nodes in an ESN (N). For a reservoir with N nodes, $\mathbf{x}(t)$ is a matrix of size $[N,1]$, \mathbf{W}_{in} is a matrix with size $[N,1]$, \mathbf{W}_{esn} is a matrix with size $[N, N]$, α is a float, f is a non-linear activation (here tanh). We note that the number of FLOPs for addition of two matrices of size $[m,n]$ is $m*n$ and the multiplication of two matrices of size $[m,n]$, $[n,p]$ is $n*m*(2p-1)$. From this we get the theoretical number of FLOPs for each subcomponent of the operation:

- $(1 - \alpha) - 1$ FLOP
- $(1 - \alpha)\mathbf{x}(t)$ – multiplication of a float and $[N,1]$ matrix = N FLOPs
- $\mathbf{W}_{in}\mathbf{s}(t)$ – multiplication of $[N,1]$ and $[1,1]$ matrix = N FLOPs
- $\mathbf{W}_{esn}\mathbf{x}(t)$ – multiplication of $[N,N]$ and $[N,1]$ matrix = $N(2N-1)$ FLOPs
- $\mathbf{W}_{in}\mathbf{s}(t) + \mathbf{W}_{esn}\mathbf{x}(t)$ – addition of two $[N,1]$ matrices = N FLOPs
- f – activation function of a $[N,1]$ matrix. Tanh takes ~ 20 FLOPs per value. = $20N$ FLOPs
- Final addition of two $[N,1]$ matrices = N FLOPs

In total, to update a reservoir we have $N + 2N-1 + N(2N-1) + N + 20N + N = 2N^2 + 24N - 3$ FLOPs. In practice, this value is higher due to the initialisation of a reservoir, and the optimisation of reservoir properties (e.g. running multiple random iterations to find an optimal configuration).

We must also take reservoir sparsity into account (echo-state networks do not have all-to-all connections, instead they are sparse networks where only ~20% of connections are non-zero) giving a final equation of $FLOPS = 2(sN)^2 + 24sN - 3$ FLOPs where s is the sparsity factor (between 0 and 1).

From this, we can benchmark the time and energy required to perform a reservoir update on a variety of different hardware. We compare the performance of FLOPS (FLOPs per second) and power consumption of GeForce RTX 4070 [NVIDIA GeForce RTX 4070 Specs | TechPowerUp GPU Database], Intel Core i7-13700H [Intel Core i713700H Processor 24M Cache up to 5.00 GHz Product Specifications], Raspberry Pi-4 [The GFLOPS/W of the various machines in the VMW Research Group (maine.edu)], and Zynq SoC Z-7020 FPGA [GPU vs FPGA Performance Comparison (bertendsp.com)]. FLOPS/Watt is calculated as $FLOPS / \text{Thermal Design Power}$.

System	FLOPS	Thermal Design Power (W)	FLOPS / Watts
GeForce RTX 4070	4.30E+13	200	2.15E+11
Intel Core i7-13700H CPU	5.38E+11	45	1.19E+10
Raspberry Pi-4	1.35E+10	6.6	2.05E+09
Zynq SoC Z-7020	1.80E+11	2.5	7.20E+10

Table 1: Performance of three conventional computing hardware options.

We can use this information to calculate the time and energy required to update a reservoir using:

$$\text{Time} = \text{FLOPs}_{\text{update}} / \text{FLOPS}$$

$$\text{Energy} = \text{FLOPs}_{\text{update}} / \text{FLOPS/Watt}.$$

Below we compare the time and energy required to update a 30 node and 500 node ESN for each of the above hardware with a sparsity of 0.2.

System	Time (30 nodes) (s)	Energy (30 nodes) (J)	Time (500 nodes) (s)	Energy (500 nodes) (J)
GeForce RTX 4070	4.95E-12	9.91E-10	5.21E-10	1.04E-07
Intel Core i7-13700H CPU	3.96E-10	1.78E-08	4.17E-08	1.87E-06
Raspberry Pi-4	1.58E-08	1.04E-07	1.66E-06	1.09E-05
Zynq SoC Z-7020	1.18E-09	2.96E-09	1.24E-07	3.11E-07

Table 2: Energy and time when updating an echo-state network with 30 and 500 nodes using conventional hardware.

We note that whilst the theoretical time taken to update a reservoir is on the order of ps – ns for some hardware, this may take longer due to the limited number of FLOPs per clock cycle.

We can compare this to the time, energy and power required to input and readout data in a nanomagnetic array. Our existing system uses a NanOsc CryoFMR PPMS probe to record spectra, and hence uses a superconducting magnet to apply fields. Operating this system carries a vast energy cost. Instead, we can compare this to a similar experimental set-up in our lab that uses an electromagnet. The electromagnet supplies a magnetic field using ~ 1 A and 5 V for 100ms, resulting in a power of 5W and energy consumption of 0.5 J. For readout, we use a microwave source with power of 17 dBm (0.05 W), which takes ~20s to record the entire spectra. This gives an energy of 1 J. Finally, the output is fed into a lock-in amplifier, which

runs at 60 W, requiring 1.2 kJ, by far the dominating factor. As such, the total power, time, and area of the existing array is 65 W, 20.1s and $\sim 1 \text{ m}^2$.

System	Power	Time (s)	Energy (J)
Field Input	1.20E+01	1.00E-01	1.20E+00
RF readout	5.00E-02	2.00E+01	1.00E+00
Lock in	6.00E+01	2.00E+01	1.20E+03

Table 3: Power, time, and energy of a comparable experimental set up.

We now calculate the power time and energy of a projected device. Below we separate out each of the components described in the manuscript. We assume Oersted field pulses of 1 ns, spin orbit torque (SOT) switching pulse of 0.25 ns, RF power of 0 dBm (1 mW) for sequential measurement (lower power RF can be used, down to -20 dBm, here we assume 0 dBm to account for circuit losses), RF noise source power of 150 mW for parallel measurement [NC520 - Noisecom | Noise Source (everythingrf.com)] and measurement time per channel of 14 ns (steady state precession requires ~ 100 oscillations which gives ~ 14 ns for a frequency of 7 GHz resonance [Ross, Andrew, et al. "Multilayer spintronic neural networks with radiofrequency connections." *Nature Nanotechnology* 18.11 (2023): 1273-1280.], for 200 channels this gives $14 \times 200 = 2800$ ns). RF source areas are based on standard surface mount component sizes. Magnetic array area is based on $5 \times 5 \text{ um}$ projected device.

System	Power (W)	Time (s)	Energy (J)	Dimensions (L x W)	Area (mm ²)
Oersted field (pulsed)	1.34E-02	1.00E-09	1.34E-11	5e-3 x 5e-3 mm	0.000025
SOT switching 2nm (pulsed)	1.70E-04	2.50E-10	4.25E-14	5e-3 x 5e-3 mm	0.000025
SOT switching 20nm (pulsed)	3.87E-01	2.50E-10	9.68E-11	5e-3 x 5e-3 mm	0.000025
Rf source (0 dBm, sequential 200 channels)	1.00E-03	2.80E-06	2.80E-09	4 x 4 mm	16
Rf source (noise source, parallel)	1.50E-01	1.40E-08	2.10E-09	4 x 4 mm	16

Table 4: Power, time, energy and area of the various components required to operate projected nanomagnetic arrays.

From this, we can calculate the power and energy of different device architectures. We start with a scheme which sequentially measures each frequency channel illustrated in Figure 1.

Figure 1: Schematic of sequential spin-torque ferromagnetic resonance readout device.

In this scheme, a sensor input in the form of a current is applied to the device via a bias tee. Following this, a broadband rf source is swept and the DC voltage output at each frequency is recorded. The power energy and time when operating this scheme is presented below in Table 5 for the two input schemes (ignoring DC amplification for now).

Input	Power (W)	Time (s)	Energy (J)
Oersted field	1.44E-02	2.80E-06	2.80E-09
SOT switching (2nm)	1.17E-03	2.80E-06	2.80E-09

Table 5: Power, time, and energy for the sequential readout scheme for Oersted field and SOT switching.

In this scheme, the time and energy for one update is dominated by the RF readout. Total power is dependent on the input scheme. This scheme requires temporary storage of each

channel in a separate memory cache which would add to the power, time, and energy. A simple, but unoptimized, way of achieving this would be to connect the hardware to a low-power microcontroller such as an Arduino Uno operating at ~100 mW.

Alternatively, we can consider the parallel readout scheme illustrated in Figure 2.

Figure 2: Schematic of the parallel readout scheme where an RF noise source excites all modes in parallel. Signal is then passed to spin torque nano-oscillator (STNO) RF-diodes and then to a memristor array for weight multiplication.

In this scheme, a sensory input is applied to the device to switch the state. An rf-noise source simultaneously excites all frequencies, and the resulting signal is sent to a series of spin-torque nano oscillator rf diodes tuned to different frequencies, serving as rf diodes. The spin-torque nano oscillators convert the rf signal into a set of DC voltages which are then passed, to a memristor array for weight multiplication. For input and readout, we obtain the following powers, energies, and update times for the two different input schemes:

Input	Power (W)	Time (s)	Energy (J)
Oersted field	1.63E-01	1.50E-08	2.11E-09
SOT switching (2nm)	1.50E-01	1.43E-08	2.10E-09

Table 6: Power, time and energy for the parallel readout scheme for Oersted field and SOT switching.

Here, the time per update has reduced by two orders of magnitude compared to the previous scheme as the readout has been parallelised. On the contrary, the power has now increased by one order of magnitude and is dominated by the rf source. The energy per update in this scheme has reduced slightly.

We must also consider the power required from any amplifiers in the circuit. In the sequential readout scheme (Figure 1) an amplifier is necessary to convert the low amplitude output voltage to a compatible read voltage for the microcontroller (we assume the sensory input is appropriately scaled to be compatible with input). The output voltage from the arrays at a power of 0 dBm is in the range of 1 – 10 uV which must be scaled to 0 – 5 V range. The required amplification to convert a 10 uV signal to 5 V is $5 / 10e-6 = 5e5$. Low power instrumental amplifiers operating at ~ 10uW powers have been demonstrated with the necessary gains [Kim, Jongpal, and Hyoungho Ko. "A dynamic instrumentation amplifier for low-power and low-noise biopotential acquisition." *Sensors* 16.3 (2016): 354.] Time and energy for this amplification would have minimal impact on the previously calculated values.

For the parallel readout scheme (Figure 3), we require a series of DC amplifiers in between the spin torque nano oscillators and the memristor array. We can use the above DC amplifiers, requiring a total power of $N \cdot 10$ uW, where N is the number of channels measured (2 mW for 200 channels). This process requires stable precession in the spin torque nano oscillator which takes 14 ns, giving an energy of $2.8e-11$ J. As such, the time per update would double to ~30 ns, whereas the energy would stay approximately the same.

In both the conventional hardware and nanomagnetic hardware, analogue to digital conversion is required. For conventional hardware, data gathered from a sensor will undergo conversion, with processing performed digitally. For nanomagnetic hardware, processing takes place in the analogue domain with readout being converted. For the sequential scheme, each frequency value will need to be converted whereas in the parallel scheme, only the readout value requires conversion. Therefore, the energy of the sequential scheme will increase as a result. Converters with high sample rate (~1 GS/s) and operating powers or 11 mW are

commonly available [Zahrai, Seyed Alireza, and Marvin Onabajo. "Review of analog-to-digital conversion characteristics and design considerations for the creation of power-efficient hybrid data converters." *Journal of Low Power Electronics and Applications* 8.2 (2018): 12.].

We can summarise the findings by comparing the four CMOS processors with the various schemes below:

Figure 3: Comparison of the power, time and energy of updating an ESN vs our projected nanomagnetic devices.

We find that the time for a single update is slower than conventional hardware for small ESNs but more comparable for large ESNs. As nanomagnetic systems are further optimised to function as well as large ESNs, then the nanomagnetic schemes will become advantageous. Our projected devices are lower energy than conventional hardware (except GPUs) even for small ESN sizes. The conventional hardware powers here are quoted as base powers, as opposed to the power required to update the ESN. Whilst our operating powers are lower in principle, there may be overheads that increase this power. Furthermore, conventional hardware can be made to operate at lower powers, but at an additional time cost. As such, this comparison serves as an indication that nanomagnetic arrays can be competitive with conventional hardware, but will be affected by how well the systems can be optimised towards their theoretical limits.

So far, we have considered single arrays. When creating a PNN using conventional hardware, the energy and time costs scale linearly with the number of echo-state networks, as adding an additional network requires the same number of FLOPs to input and update. The total power remains the same.

For the nanomagnetic arrays, adding additional reservoirs to the sequential scheme would require a relay to control where the input signal is sent at any given time Figure 4. This could, in principle, be achieved with transistors or MEMS components. The RF signals can also be sent to certain locations via rf-mems components. These components require ~0.1 mW to operate and hence do not greatly affect the overall power consumption of the proposed devices.

Figure 4: Example schematic of a hardware PNN with sequential rf readout method.

In this case, the time and energy would be a factor of N larger, where N is the number of magnetic arrays. Here, the area of the device remains dominated rf source.

For the parallel readout scheme, each array would need to be sequentially measured to receive input from the previous array in the network. As such, we again find that the time and energy would increase by a factor of N for this system.

In summary, in both software and hardware options, the energy and time scales in the same manner with increasing PNN size, therefore the previous comparison of performance and energy with a single array is sufficient for comparison.

Whilst the focus here is on nanomagnetic reservoirs, our PNN scheme can be used with other low-power / low-energy neuromorphic technologies. One example is low power memristor arrays which run at ~ 22 μW [Zhong, Yanan, et al. "A memristor-based analogue reservoir computing system for real-time and power-efficient signal processing." *Nature Electronics* 5.10 (2022): 672-681.]. If one were to interconnect these dynamic memristors into PNNs, one can benefit from the computational advantages of the networked approach, at a fraction of the power consumption of our proposed devices.

Second, the second comment was also not addressed at the hardware level. They should not only use MLP. They should build faithful reservoirs that are similar to PNN using CMOS and then benchmark the resources.

After seeking clarification about the above request, we received the following information:

"I suggested that they should initialize and interconnect multiple software reservoirs (like ASIs) together in a combined parallel/deep architecture corresponding to the PNN. This enables them to compare MSE. Then, they should design the physical CMOS hardware reservoirs using simulations that are efficient in implementing the software reservoirs. This enables them to benchmark the power, energy, and speed."

We thank the reviewer for the suggestion. In response, we have conducted a comparison involving three echo-state networks (ESNs) with characteristics similar to the experimental arrays. These ESNs were interconnected using the same methodology as applied to the nanomagnetic arrays.

We begin by initializing ESNs with 200 nodes (to ensure a similar number of output channels in comparison to nanomagnetic arrays) and select three which display similar memory-capacity and non-linearity to the three nanomagnetic arrays. Figure 6 shows the memory-capacity and non-linearity of the three nanomagnetic arrays alongside three ESNs with similar properties (red circles). The ESNs were found via randomly varying the hyperparameters of the ESN and selecting ESNs which had the closest characteristics.

Figure 5: a) Memory-capacity (MC) and non-linearity (NL) of the nanomagnetic arrays and three ESNs (red markers) chosen to have similar metrics. b) Performance when predicting future values of the Mackey-Glass equation for nanomagnetic arrays (solid line) and ESNs with similar metrics (dotted lines).

We find that, despite the ESNs and nanomagnetic arrays displaying similar metrics, the performance shows different characteristics. For the two high memory-capacity ESNs, the MSE rises slightly from future step 1 – 3, and then becomes flat, whereas the hardware reservoirs display a periodic profile in the MSE. We find that it is challenging to find an ESN that displays the same characteristics as the hardware reservoirs. This is not surprising, as nanomagnetic reservoirs and software ESNs are governed by different underlying dynamics. In nanomagnetic arrays, coupling arises from the dipolar field which decays spatially and hence coupling only occurs locally, whereas in ESNs, coupling is randomly assigned across the network. In ESNs, the coupling strengths between nodes are initialised and remain fixed, whereas in nanomagnetic arrays, the coupling between neighbouring elements depends on the state of those elements (e.g. uniform magnetisation or vortex). Furthermore, in ESNs, only certain nodes are coupled to the input, which enhances memory-capacity as certain nodes require a number of network updates before receiving information about a particular input. On the other hand, in nanomagnetic arrays, each node is subject to the input data when it is first applied. Finally, experimental nanomagnetic systems are subject to noise, whereas ESNs are noise free. It is possible that, with extensive optimisation, an ESN could be initialised that has more similar characteristics to nanomagnetic arrays, however, achieving this is beyond the scope of this report.

We note that when initialising ESNs, the majority of them had memory-capacity far beyond the nanomagnetic arrays (close to 8).

We now interconnect these arrays following the same methodology used to connect different nanomagnetic arrays. We begin by analysing the channel specific time correlation and non-linearity of the three ESNs displayed in Figure 7.

Figure 6: Per-channel correlation and non-linearity for the three selected ESNs.

When comparing these results to the nanomagnetic reservoirs (SI Figure 5) we again see fundamental differences in the dynamics of ESNs vs nanomagnetic hardware. For the low memory-capacity ESN (ESN 1), we find that all channels are only correlated with short-term previous inputs. Conversely, for the higher memory-capacity ESNs (ESN 2&3), there are a broad range of correlations. Some channels are correlated with short term previous inputs, whereas some are correlated with previous inputs many time-steps ago. In contrast, in nanomagnetic arrays, all of channels are most strongly correlated with short-term inputs and only a handful are correlated with long-term previous inputs. This is a symptom of the differences between internode couplings in ESN when compared to nanomagnetic arrays. The random, sparse connections in ESN allow many different dynamic timescales whereas the spatially constricted, local couplings in nanomagnetic systems prevent this behaviour.

For each ESN, we now select six channels: three with the highest memory capacity and three with the highest non-linearity, giving a total of 18 input time-series to feed into the next layer. Each ESN is the subject to all input sequences to produce 54 2-series networks. We then create 3-series networks by selecting the 2-series architectures which: 1) begin with the low memory-capacity reservoir (ESN 1) and 2) which display the lowest MSE. We then perform the same per-channel metric analysis, generating 9 input sequences which are passed to the final ESN to form a 3-series network. Finally, we combine the responses from all single, 2-series and 3-series networks using the same methodology as the PNN. We evaluate the performance of the ESN networks using the same feature selection methodology described in the Methods section of the manuscript.

Figure 7: Comparison between nanomagnetic arrays and software ESNs when interconnecting networks. Top row shows the experimental arrays from the main manuscript. The bottom row shows the ESN networks.

Figure 8 compares the performance of single, 2-series, 3-series and PNN architectures for the three prediction tasks presented in the paper. The top row shows the nanomagnet response previously included in the manuscript and the bottom row shows the ESN network results. The ESN results displayed are taken from the architecture with the lowest MSE over all future / NARMA steps for a given task. When interconnecting ESNs, we find that whilst improvements are observed for some tasks, they are less pronounced than for the physical system. Series networks only show improvements for future prediction $t+1$ and $t+2$, NARMA 10 and above,

and NARMA + Future prediction t+4 to t+13, whereas series networks improve all predictions for the nanomagnetic arrays. Software analogues of the PNN architecture show improvements for more tasks (notable future prediction t+7 – t+12), but also have worse performance for other tasks. We believe this is because single ESNs already display a rich set of memory timescales, and hence interconnecting arrays does not enrich the overall network to the same extent. These results give us an indication that interconnecting physical systems can give MSE improvements which produce a similar level of performance to software counterparts. However, we stress here that the initial ESNs selected are not optimised for performance. If we were to optimise ESNs for performance, then it is likely that the software network would outperform nanomagnetic hardware.

In addition, we test deep echo state networks, where multiple interconnections between different ESN layers are made (i.e. multiple nodes from one ESN connect to multiple nodes in another) in a similar manner to previous work of some of our authors [Manneschi, Luca, et al. "Exploiting multiple timescales in hierarchical echo state networks." *Frontiers in Applied Mathematics and Statistics* 6 (2021): 616658.]. We connect the three ESNs with similar characteristics to the physical nanomagnetic reservoirs into a 3 x 3 network, whereby each ESN is present in each layer. Layer interconnections are initialised randomly and we average over 10 random trials. The prediction results for the three tasks are shown in Figure 8 (red curve, 'Deep ESN'). As expected, the performance dramatically improves compared to our proposed method as there is an increase in the amount of information being transferred between different layers. Currently, such interconnections are not possible with the hardware approach as each input is only able to accept one input per time step. If a physical system is used which can take multiple inputs at each time step (e.g. memristors), we can connect it in this way and achieve a computationally powerful network with just a handful of nodes.

We once again thank the reviewer for their suggestions for improving the comparison to software. We have incorporated these results into the supplementary information of the manuscript.

We appreciate the suggestion to perform a concrete comparison with existing technology to validate the technological viability of new computing hardware. However, the primary focus of this manuscript is not to assert that nanomagnetic arrays are a technologically viable substrate for computing. Instead, our objectives are twofold: a) to demonstrate that the performance of physical reservoir computing can be enhanced through our proposed interconnection methodology, and b) to show that a physical system/network can achieve an overparameterised state, thereby enabling few-shot learning.

Given these specific aims, conducting a full simulation of the CMOS architecture required to run an echo-state network would exceed the intended scope of this study. We have, however, calculated the theoretical minimum number of operations needed to update a software echo-state network and have also provided calculations of the energy and time requirements when performing these operations on various CMOS-based hardware systems. We trust that this additional analysis sufficiently facilitates comparisons with existing hardware systems and aligns with the focus of our paper. We hope that the reviewers find these contributions to be a compelling enhancement to the discourse on the capabilities of advanced computing architectures.

Third, I am not in agreement with a statement like this: "To be clear, none of our results are purely software simulation. All the data is recorded experimentally, only the training of weights takes place offline." In my opinion, the spectra of these ASIs were only taken at a particular time. Then, the authors assumed that these data could be repeatedly used for their "simulation."

However, it is not a real experiment due to the following reasons. (1) The spectra can drift significantly over time! So, the data taken at time 0 can not be used for simulation at time T without justification.

(2) In all simulation experiments (like running a PNN to achieve over-parametrization), one needs to repeatedly measure the PNN spectra, and thus, actual measurements on the PNN will take a huge amount of time, which is also feasible to me.

The reviewer is correct that the real measurement times do not correspond to simulation time. We apologise for the previous misunderstanding. To create a series network, we first measure the entire spectra from one network. We then select one FMR channel output to serve as the input to the next array. The next array is loaded into the experimental set up and then measured. Finally, we concatenate the spectra. As such, the reviewer is correct that results measured at different real times are shifted to become the same time when assessing the PNN performance. This is necessary to evaluate the PNN performance without needing multiple experimental set-ups.

The reviewer raises an excellent point regarding drift. In many physical systems, spectral drift occurs due to a variety of reasons. Spin-wave measurements are typically robust against drift. We have tested the experimental drift in our system by recording the spectra in a -18 mT bias field ~2200 times of 24 hours. Below we plot the spectra of the WM array, both raw signal and the signal after subtracting the mean of all spectra. We find there is no drift in our experimental set up. In a future device, this would remain true as the arrays are fabricated in direct contact with waveguides or heavy metals.

Figure 8: FMR signal as a function of time, measured over 24 hours. The signal shows only small amounts of drift around 7.7 GHz at amplitudes ~40 X less than the main signal.

Furthermore, in the overparameterised regime we have shown that the system is able to rapidly adapt to new tasks. This is like operating in a new environment (ie temperature variation) hence we expect the network to be robust against temperature induced drifts. In practice, one could periodically retrain the array to account for any drift in spectra.

With regards to measurement time, the reviewer raises a good point about the existing scheme in that we are replicating a PNN using three arrays, where each array serves as multiple nodes within the PNN. The reviewer is correct that if one were to use this method in practice, each time a new input was received, one would need to repeat a portion of the previous inputs to ensure that the history-dependent response remains the same.

In practical applications for future devices, we envisage fabricating multiple arrays as described in our energy analysis. Each array would function as a single node in the network, with inputs applied in real time. Each node would retain its state until the next input cycle. Consequently, the time required to measure the entire PNN would scale linearly with the number of nodes, as previously described. We have included this information in the new discussion section. Furthermore, we underline that our focus in this paper is demonstrating the conceptual advances that interconnecting networks of physical reservoirs provides,

mirroring the computational benefits observed in interconnected software-based ESNs/reservoirs.

Reviewer #3 (Remarks to the Author):

The authors have provided a comprehensive response to the questions raised. My only remaining concern is the broad use of "neuromorphic computing". While it is appropriately replaced by "physical reservoir computing" in most instances, I disagree with its use in the phrase "we present solutions to key outstanding problems in the physical neuromorphic computing field". As mentioned in my initial review, the field of physical neuromorphic computing is much broader than just reservoir computing, and the problems targeted by the authors do not exist in many of its sub-areas.

Once this is addressed, the paper can be recommended for publication.

p.s. there is a typo in: "show allow memory and non-linearity improvements"

We thank the reviewer for their positive appraisal and the opportunity to improve our manuscript. We have addressed your points and implemented the suggested changes in the manuscript.

REVIEWER COMMENTS

Reviewer #3 (Remarks to the Author):

The raised concerns have been addressed, and the clarity of the manuscript has improved. The results are novel and interesting. I thus recommend this work for publication.

Reviewer #3 (Remarks to the Author):

The raised concerns have been addressed, and the clarity of the manuscript has improved. The results are novel and interesting. I thus recommend this work for publication.

We thank the reviewer for their positive appraisal.

REVIEWERS' COMMENTS

Reviewer #3 (Remarks to the Author):

I am satisfied with the revisions made and have no further comments or suggestions.